

# Proximity-induced gapless superconductivity in two-dimensional Rashba semiconductor in magnetic field

Serafim S. Babkin[1], Andrew P. Higginbotham[1,2] and Maksym Serbyn[1]

**1** Institute of Science and Technology Austria (ISTA),
Am Campus 1, 3400 Klosterneuburg, Austria
**2** The James Franck Institute and Department of Physics,
University of Chicago, Chicago, Illinois 60637, USA

## Abstract

Two-dimensional semiconductor-superconductor heterostructures form the foundation of numerous nanoscale physical systems. However, measuring the properties of such heterostructures, and characterizing the semiconductor *in-situ* is challenging. A recent experimental study by [1] was able to probe the semiconductor within the heterostructure using microwave measurements of the superfluid density. This work revealed a rapid depletion of superfluid density in semiconductor, caused by the in-plane magnetic field which in presence of spin-orbit coupling creates so-called Bogoliubov Fermi surfaces. The experimental work used a simplified theoretical model that neglected the presence of non-magnetic disorder in the semiconductor, hence describing the data only qualitatively. Motivated by experiments, we introduce a theoretical model describing a *disordered* semiconductor with strong spin-orbit coupling that is proximitized by a superconductor. Our model provides specific predictions for the density of states and superfluid density. Presence of disorder leads to the emergence of a gapless superconducting phase, that may be viewed as a manifestation of Bogoliubov Fermi surface. When applied to real experimental data, our model showcases excellent quantitative agreement, enabling the extraction of material parameters such as mean free path and mobility, and estimating $g$-tensor after taking into account the orbital contribution of magnetic field. Our model can be used to probe *in-situ* parameters of other superconductor-semiconductor heterostructures and can be further extended to give access to transport properties.



# 1 Introduction

Two-dimensional superconductor-semiconductor heterostructures, where the semiconductor possesses Rashba spin-orbit coupling (SOC) [2], have attracted a lot of attention recently due to several promising applications. These include superconducting qubits [3,4], engineered $p$-wave superconductivity [5,6], and, when subjected to a magnetic field, they present a promising platform for hosting Majorana zero-modes [7–10]. However, experimental investigations of such heterostructures pose challenges. In transport measurements, for instance, the superconductor acts as a shunt. As a result, in order to study properties of the buried interface, some additional experimental probes are required. To address this requirement, recent experiments have probed superfluid density [1,11], vortex inductance [12], and terahertz cyclotron resonance [13].

    The measurement of the superfluid density in superconductor-semiconductor heterostructure made out of aluminum deposited on top of spin-orbit coupled two dimensional electron gas (2DEG) using a resonant microwave circuit was implemented by some of the present authors and collaborators [1]. The key insight used in the experiment [1] was the qualitatively different response of superfluid density in the conventional superconductor (SC) and proximitized 2DEG to in-plane magnetic field. Specifically, at a certain value of magnetic field Bogoliubov bands in 2DEG touch the Fermi energy, causing an abrupt depletion of superfluid density that was confirmed by the experiment.

Although the superfluid density depletion observed in experiment confirmed the proximity-induced order parameter in spin orbit coupled 2DEG, numerous questions remained open. In particular, to extract specific parameters of 2DEG from experimental data, $g$-factor, mobility, or carrier density, Ref. [1] relied on the simplified theoretical model that neglected disorder in 2DEG. This model resulted only in a qualitative agreement with the data, thus calling for more realistic theoretical model. Another outstanding question was the fate of the system in the regime when Bogoliubov bands touch the Fermi energy. On the one hand, at this point so-called Bogoliubov Fermi surfaces were predicted to emerge [6]. On the other hand, in disorder-free theories the superfluid density contribution from 2DEG becomes negative in that regime, signaling potential instability, that was previously considered in a number of different contexts [14–16]. Thus, reliable theoretical description of the system in the regime with Bogoliubov Fermi surfaces remains missing.

In this paper we develop a more realistic model for the proximitized 2DEG with strong spin-orbit coupling that incorporates the presence of non-magnetic disorder and in-plane magnetic field. To describe the proximitized 2DEG with disorder, we use a Green's function formalism and perform a self-consistent calculation of the self-energy in the 2DEG, that takes into account impurity scattering. Using Green's function, we calculate the density of states (DOS) in the 2DEG and its superfluid density. Our main finding is that disorder leads to a regime of stable gapless superconductivity for a range of magnetic fields that expands with disorder strength. Although in the presence of impurities momentum is not a good quantum number, the gapless regime in presence of disorder may be viewed as a stable generalization of the state with Bogoliubov Fermi surfaces to the case of a disordered material.

We note that the 2DEG with spin-orbit coupling and pairing was previously considered in the literature. In particular, in early works [17, 18], the 2D superconductor with spin-orbit coupling was studied. In these papers, in contrast to ours, superconductivity is an internal property of 2D layers and does not come from an external superconductor through the proximity effect. Besides, in Ref. [17] the non-magnetic disorder was not incorporated. Later works [5–8] studied the superconductor-2DEG heterostructure without incorporating non-magnetic disorder and suppression of order parameter by magnetic field. Several studies have incorporated the effects of disorder and band-bending to more realistically model the superconductor-semiconductor interface [19–23]. Also, proximity induced triplet pairing was considered in Ref. [24] but for low carrier concentrations when spin-orbit splitting and Fermi energies are comparable.

In addition to 2DEG proximitized by conventional s-wave superconductors, the physics of Bogoliubov Fermi surfaces and interplay between disorder, spin-orbit coupling and superconductivity is actively studied in other material systems. In particular, Bogoliubov Fermi surfaces in presence of in-plane magnetic field were probed by the scanning tunneling microscopy experiments performed on the proximitized surface states of topological insulator [25]. Also, Bogoliubov Fermi surfaces were studied in the multiband superconductors with broken time-reversal symmetry [26–30]. However, typically in such systems the pairing has higher angular momentum and is expected to be rapidly suppressed by disorder. In a different direction, Refs. [31–35] considered the phase diagram and effect of disorder and magnetic field on the so-called Ising superconductivity, where strong spin-orbit coupling locks spin to a particular direction.

Application of our theoretical framework to the real experimental data from Ref. [1] results in a much better quantitative agreement with the data, compared to the oversimplified theoretical model in [1]. Using simultaneous multi-fitting of the model parameters, we extract the values of scattering time, mobility, and carrier density of the 2DEG, and estimate values of the $g$-tensor anisotropy and $g$-factor. The mobility is qualitatively consistent with Hall measurements [1], and extracted scattering time suggests that semiconductor has disor-

der strength that puts it in between clean and dirty regimes. Moreover, we identify the fairly broad range of magnetic fields where the 2DEG realizes gapless proximity-induced superconductivity, and predict the shape of the density of states that could be potentially probed in tunneling experiments.

Although our model results in a quantitative agreement with experimental data, it still has a number of limitations that are related to our treatment of proximity effect. Specifically, we ignore an inverse proximity effect, and also treat the orbital effect of the magnetic field related to the motion of carriers between superconductor and 2DEG only qualitatively. Incorporation of these effects is important for understanding the ground state of the heterostructure from Ref. [1] for magnetic fields larger than 0.75 T, where our model still predicts that the superfluid turns negative signaling a potential instability. Also, self-consistent treatment of orbital effect of magnetic field is required for a more reliable extraction of the $g$-factor, that is currently done using phenomenological considerations. We hope that these shortcomings of our model can be addressed in the future work. The present theoretical model may be used as a guide for the values of magnetic field required for realizing the potential instability of the gapless superconducting state and studying it in future experiments.

More broadly, our theoretical model with some modifications will be applicable to a much broader family of materials, such as hybrid semiconductor [36] and topological insulator [37, 38] nanowires or proximitized surface states [25], Germanium based 2DEGs [39], ferromagnetic hybrids [11], and two-dimensional materials with strong spin orbit coupling such as transition metal dichalcogenides. Also, it can be extended to predict the dissipative electromagnetic response, spin susceptibility and other characteristics accessible in the future experiments. Such extension of our work could allow a comprehensive *in-situ* characterization of heterostructures before fabrication of more complicated devices that i.e. attempts to realize Majorana physics [9, 10].

The rest of the paper is structured as follows: In Section 2 we introduce a theoretical model based on the Green's function formalism. This section covers the calculation of the density of states (DOS) and superfluid density using Green functions. We also discuss how the magnetic field affects the order parameter and extend our model to consider cases with magnetic anisotropy. Next, in Section 3 we discuss theoretical predictions for the DOS and superfluid density, both with and without suppression of the order parameter by the magnetic field. In addition, we identify the parameter range of magnetic field where the semiconductor has gapless proximity-induced superconductivity as a function of disorder strengths. In Section 4 we introduce and implement a fitting procedure for experimental data for Al-InAs heterostructure [1], discuss the resulting material parameters and show predictions for the density of states. Finally, we conclude in Section 5 with the summary of our results and outstanding open questions.

## 2 Theoretical model

In this section we introduce the main theoretical model for describing the 2DEG proximitized with a conventional superconductor. We begin by describing the physical system and the approximations used in our theoretical treatment, covered in Sec. 2.1 and Sec. 2.2. Following this, the Green's function formalism is introduced in Sec. 2.3, where we also discuss self-consistency conditions. The derivation of physical properties, such as the density of states and superfluid density, is detailed in Sec. 2.4. Then in Sec. 2.5 we discuss the generalization of our calculations to the case of anisotropic $g$-factor. Finally, we conclude this section with the review of the depairing theory of the conventional superconductor in Sec. 2.6 required for the realistic modeling of the heterostructure.

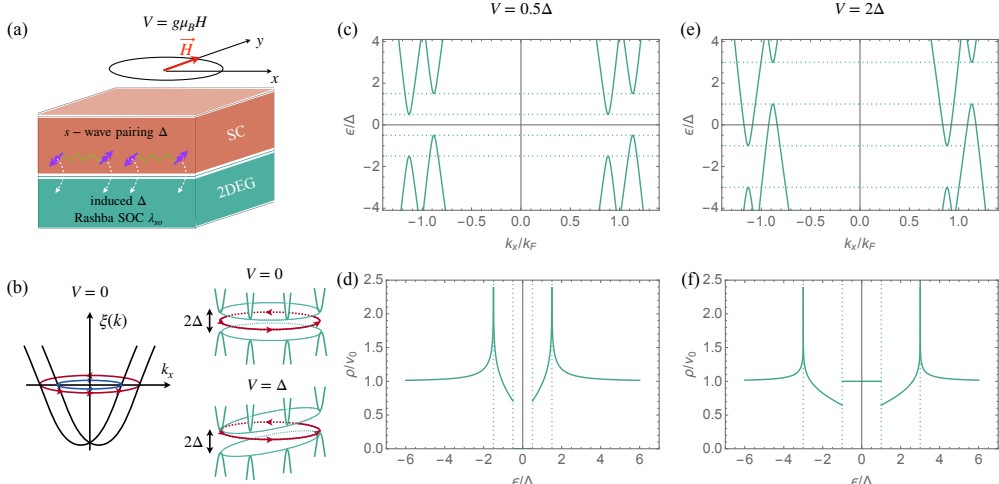

Figure 1: (a) Sketch showing 2D-heterostructure, where the superconductor (SC) with the s-wave pairing proximitizes the 2DEG. The 2DEG has strong Rashba spin-orbit coupling (SOC), and the entire heterostructure is subject to the in-plane magnetic field pointing in $y$-direction. (b) Rashba-split bands of 2DEG spectrum without magnetic field lead to two spin-momentum locked Fermi surfaces (red and blue) shown on the left. The right part of panel (b) shows the red Fermi surface gapped due to proximity effect without (top right) and with (bottom right) the magnetic field. The presence of a magnetic field tilts the spectrum in an anisotropic manner with gapped bands touching the Fermi level first along the $x$-direction. (c) Cut of the band structure in presence of induced gap and weak magnetic field $V < \Delta$ along the $x$-axis and (d) associated DOS that shows the decreased gap, the discontinuous jump in DOS at the gap edge, followed by the logarithmic Van Hove singularity. Panels (e)-(f) show similar data for larger value of magnetic field $V > \Delta$, when Bogoliubov's Fermi surfaces form and DOS becomes gapless. Panels (c) and (e) use chemical potential $\mu = 20\Delta$ and spin orbit coupling $\lambda_{so}k_F = 5\Delta$ in units of proximity-induced gap, $\Delta$.

## 2.1 System and physical assumptions

We study 2D heterostructure which consists of superconducting and semiconducting layers schematically depicted in Fig. 1(a). The superconductor is assumed to be isotropic material with a conventional s-wave order parameter. Moreover, motivated by the common use of aluminum, we assume that superconductor is strongly disordered. The 2DEG layer, in contrast is generally anisotropic and has a strong spin-orbit coupling. For simplicity, we choose the spin-orbit coupling to be of the Rashba form with potential generalizations discussed in Appendix D. Moreover, we also neglect the anisotropy in most of 2DEG parameters, and take into account only anisotropic $g$-factor that controls the coupling to the external in-plane magnetic field of varying direction and magnitude, see Fig. 1(a). Finally, the new crucial ingredient incorporated in the present work is the presence of disorder of arbitrary strength in the 2DEG layer.

Simultaneous presence of spin orbit coupling and magnetic field breaks both spin-rotation symmetry and time-reversal symmetries of the system. The effect of the in-plane magnetic field on the conventional superconductor is the suppression of the order parameter, that can be described by the conventional depairing theory [40, 41] and will be reviewed in Sec. 2.6. Likewise, we incorporate the coupling of the in-plane magnetic field to the charge carriers in the 2DEG. However, assuming the small thickness of the 2DEG, we take into account only Zeeman term [42].

The crucial assumption that underlies our treatment is that the 2DEG gets the superconducting order parameter by proximity effect but is not affecting the order parameter in the superconductor. Such absence of inverse proximity effect from the 2DEG onto superconductor (that would result in additional suppression of the order parameter in the superconductor) is justified if the density of carriers is much higher in the superconducting material. The absence of inverse proximity effect along with the assumption of transparent interface between semiconductor and superconductor (we note that our model can be easily extended to include suppression of the induced order parameter due to imperfect interface transparency, in practice it is also possible to characterize induced order parameter in the 2DEG via tunneling measurements) allows us to treat the 2DEG with the induced order parameter that is entirely determined by depairing intrinsic to superconductor. This approximation distinguishes our model from earlier Ref. [18], where the pairing mechanism was assumed to be intrinsic to the 2DEG.

## 2.2 Hamiltonian and band structure

In order to construct the Green's function, we first consider the system in the absence of disorder in the momentum space. To this end, we use the enlarged space that includes spin and Nambu (particle-hole) degrees of freedom, with operators $c_{\mathbf{k}}^{\dagger}(c_{\mathbf{k}})$ that are Fourier transformation of real space creation (annihilation) operators, $\psi^{\dagger}(\mathbf{r})(\psi(\mathbf{r}))$, $C_{\mathbf{k}}^{\dagger} = [c_{\mathbf{k}\uparrow}^{\dagger}, c_{\mathbf{k}\downarrow}^{\dagger}, c_{-\mathbf{k}\uparrow}, c_{-\mathbf{k}\downarrow}]$. We write the Hamiltonian as:

$$\mathcal{H}_0 = \int \frac{d^2\mathbf{k}}{(2\pi)^2} C_{\mathbf{k}}^{\dagger} h_0(\mathbf{k}) C_{\mathbf{k}}, \tag{1}$$

$$h_0(\mathbf{k}) = \tau^z \xi_{\mathbf{k}} + \lambda_{\mathrm{so}} k_F [\tau^z \sigma^y \cos\phi_{\mathbf{k}} - \sigma^x \sin\phi_{\mathbf{k}}] - \sigma^y V - \tau^y \sigma^y \Delta, \tag{2}$$

where the function $\xi_{\mathbf{k}} = k^2/(2m) - \mu$ encodes the parabolic band structure with effective mass $m$ and chemical potential $\mu$. Parameter $\lambda_{\mathrm{so}}$ determines the strength of the Rashba spin-orbit coupling, and we used approximation $\lambda_{\mathrm{so}} k \approx \lambda_{\mathrm{so}} k_F$ ($k_F$ is a Fermi momentum), that assumes that spin orbit splitting is much smaller than the Fermi energy. Set of Pauli matrices $\sigma^{x,y,z}$ acts in the spin space and Pauli matrices $\tau^{x,y,z}$ operate in the particle-hole space. We introduced $\phi_{\mathbf{k}}$ as the angle of momentum direction with respect to $x$-axis, $\cos\phi_{\mathbf{k}} = k_x/k$. Finally, terms in the second line of Eq. (2) describe the Zeeman energy and the proximity-induced isotropic order parameter.

In the definition of Hamiltonian (2) we fix the direction of magnetic field to point along $y$-axis as in Fig. 1(a). This results in the Zeeman energy $-\sigma^y V$ with

$$V = \frac{1}{2} g \mu_B H, \tag{3}$$

where $g$ is the isotropic $g$-factor (generalization to anisotropic $g$-tensor is presented in Sec. 2.5 below) and $\mu_B$ is the Bohr magneton. We emphasize the presence of a factor $1/2$ in Eq. (3) that was omitted in Refs. [1, 6] but is essential for comparing the values of $g$-factor with the previous literature. Another consequence of the in-plane magnetic field, not included in Eq. (2), is its orbital effect, which is known to introduce an additional phase between carriers in the superconductor and 2DEG [43]. While incorporating this effect into our self-consistent treatment is beyond the scope of the present work, we take it into account phenomenologically in Sec. 4.3 in order to extract the realistic values of $g$-factor.

The left panel of Figure 1(b) shows a sketch of the spin-orbit locked Fermi surfaces corresponding to the Hamiltonian (2) without pairing and magnetic field. Non-zero induced order parameter, $\Delta$, without magnetic field $V = 0$ results in the isotropic gap opening for each of the two spin-momentum locked Fermi surfaces, see the sketch in Fig 1(b) on the right. Non-zero

magnetic field, $V > 0$, results in the tilting of the Bogoliubov's bands. Although formally the distance between top and bottom of quasiparticle band remains $2\Delta$ (for now we neglect the depairing effect of magnetic field on superconductor), the separation from the hole branch and Fermi level decreases for $k_x > 0$ and increases for the negative $k_x < 0$, see Fig 1(b). At the special value of the field, when $V = g\mu_B H/2 = \Delta$, the Bogoliubov's bands touch the Fermi level, signaling emergence of Bogoliubov's Fermi surfaces [6].

The qualitatively different form of the band structure and associated density of states (DOS) is illustrated in Fig. 1(c)-(d) for $V < \Delta$ and in panels (e)-(f) when $V > \Delta$. In the latter case, the DOS has no gap anymore — a direct manifestation of the emergence of Bogoliubov's Fermi surfaces. We note, that although the analytic expression for the band structure resulting from Eq. (2) was obtained in the previous literature [6], the associated DOS was studied only numerically. In the Appendix A we present the analytic derivation of the DOS for this band structure. We show that the standard square-root singularity in the DOS at $\Delta$ is split due to presence of magnetic-field and spin-orbit coupling into two Van Hove singularities: at the band bottom the DOS has a discontinuous jump at energies $\pm|\Delta - V|$ rather than square root divergence as in BCS case. In addition, at energies $\pm|\Delta + V|$ the band structure develops a logarithmic Van Hove singularity, see Fig. 1(d) and (f).

We are interested in the case when the energy scale associated with the spin-orbit coupling much larger compared to Zeeman energy, $\lambda_{so} k_F \gg V$. In this case, the band-splitting due to spin-orbit coupling is much larger compared to the Zeeman-induced anisotropic shift of energy bands. This allows us to neglect the inter-band terms induced by the magnetic field, as they are suppressed by the small parameter $[V/(\lambda_{so} k_F)]^2$. More details on this approximation are presented in the Appendix A.

## 2.3 Green function formalism in presence of disorder

We incorporate the non-magnetic disorder using the self-energy in the Green's function formalism. The matrix form of disorder potential in coordinate representation reads $h_{dis}(\boldsymbol{r}) = \tau^z U(\boldsymbol{r})$, where $U(\boldsymbol{r}) = \sum_i u_0 \delta(\boldsymbol{r} - \boldsymbol{r}_i)$ and $u_0$ and $\boldsymbol{r}_i$ are the impurity strength and coordinate location respectively. We calculate the average self-energy $\Sigma$ due to scattering of electrons by the impurity potential,

$$\Sigma = n_{imp} u_0^2 \tau^z \int_{\boldsymbol{k}} G(i\epsilon_n, \boldsymbol{k}) \tau^z, \tag{4}$$

here and in what follows we use the following notation: $\int_{\boldsymbol{k}} \ldots = \int \frac{d\boldsymbol{k}}{(2\pi)^2} \ldots$, and denote by $n_{imp}$ the concentration of non-magnetic impurities. The average self-energy enters into the expression for the Greens function in presence of disorder,

$$G^{-1} = i\epsilon_n - h_0(\mathbf{k}) - \Sigma, \tag{5}$$

where the first two terms correspond to the inverse of the Green's function without disorder defined with Hamiltonian (2), and $\epsilon_n = 2\pi T(n + 1/2)$ correspond to fermionic Matsubara frequencies.

The equations (4)-(5) should be solved in a self-consistent manner which leads to the renormalization of parameters in Green's function [44–46]. Specifically, this results in the following form for self-energy:

$$\Sigma = i(\epsilon_n - \epsilon^{(1)}) + \sigma^y(V - V^{(1)}) - \tau^y \sigma^y(\Delta - \Delta^{(1)}) + \tau^y \Delta^{(2)}. \tag{6}$$

Physically, this equation implies that disorder scattering renormalizes all parameters present in the original Green's function, such as gap, quasiparticle residue [44, 45], and Zeeman energy. Moreover, in addition to singlet pairing, the disorder scattering induces the odd-frequency

triplet component of the order parameter encoded by the function $\Delta^{(2)}$ [46]. Substituting self-energy (6) into Eq. (5) we set the explicit expression for the Green function,

$$G^{-1} = i\epsilon^{(1)} - \tau^z \xi_{\mathbf{k}} - \lambda_{\mathrm{so}} k_F [\tau^z \sigma^y \cos\phi_{\mathbf{k}} - \sigma^x \sin\phi_{\mathbf{k}}] + \sigma^y V^{(1)} + \tau^y \sigma^y \Delta^{(1)} - \tau^y \Delta^{(2)}, \quad (7)$$

where original parameters are replaced by their renormalized versions and also induced triplet-paring order parameter component, $\Delta^{(2)}$, appears. In what follows we neglect interband terms in the Green function which amounts to neglecting contributions of the order of $[V^{(1)}/(\lambda_{\mathrm{so}} k_F)]^2$ and $[\Delta^{(2)}/(\lambda_{\mathrm{so}} k_F)]^2$, see Appendix B. Finally, we emphasize that all four renormalized parameters are non-trivial functions of frequency, $\epsilon_n$, unlike their bare counterparts.

Substituting self-energy (6) into the self-consistent equation (4) yields four self-consistent conditions for each of the renormalized parameters,

$$\begin{cases} i\epsilon^{(1)} - i\epsilon_n = \left\langle i\epsilon^{(1)} - V^{(1)} \cos\phi_{\mathbf{k}} \right\rangle_{\mathbf{k}}, \\ \Delta^{(1)} - \Delta = \left\langle \Delta^{(1)} + \Delta^{(2)} \cos\phi_{\mathbf{k}} \right\rangle_{\mathbf{k}}, \\ V - V^{(1)} = \left\langle \cos\phi_{\mathbf{k}} \left( i\epsilon^{(1)} - V^{(1)} \cos\phi_{\mathbf{k}} \right) \right\rangle_{\mathbf{k}}, \\ \Delta^{(2)} = \left\langle \cos\phi_{\mathbf{k}} \left( \Delta^{(1)} + \Delta^{(2)} \cos\phi_{\mathbf{k}} \right) \right\rangle_{\mathbf{k}}, \end{cases} \quad (8)$$

where we introduced the following notation:

$$\left\langle O \right\rangle_{\mathbf{k}} = \frac{1}{2\pi\tau} \int d\xi \int_0^{2\pi} \frac{d\phi_{\mathbf{k}}}{2\pi} \frac{O}{\xi^2 + (\Delta^{(1)} + \Delta^{(2)} \cos\phi_{\mathbf{k}})^2 - (i\epsilon^{(1)} - V^{(1)} \cos\phi_{\mathbf{k}})^2}, \quad (9)$$

for the integral over momentum magnitude and direction. Here the mean scattering time $\tau$ is defined as $\tau = (2\pi n_{\mathrm{imp}} u_0^2 \nu_0)^{-1}$ with $\nu_0$ being the density of states per spin projection.

The solutions to the self-consistent system of equations (8) generally cannot be found analytically. In the Section 3 we discuss the details of numerical scheme that is used for solving such equations. Below, we assume that the equations are solved and discuss the calculation of the physical observables from the resulting self-consistent parameters $\epsilon^{(1)}, V^{(1)}, \Delta^{(1)}, \Delta^{(2)}$ that completely determine the Green's function of 2DEG in presence of disorder.

## 2.4 Density of states and superfluid density

The DOS can be calculated using its expression via Green function and explicit parametrization of the latter in Eq. (7):

$$\rho(\epsilon) = -\frac{1}{4\pi} \int_{\mathbf{k}} \mathrm{Im}\,\mathrm{Tr}\,G(\epsilon + i0, \mathbf{k}) = 2\tau\nu_0 \mathrm{Im}\,\epsilon^{(1)}(\epsilon), \quad (10)$$

where $\tau$ is the mean scattering time and $\nu_0$ is the density of states at each of the spin-orbit split Fermi surfaces. Here we used the retarded Green function which is analytical continuation of Matsubara Green function $G$ from imaginary frequencies to real ones: $i\epsilon_n \to \epsilon + i0$. Analogously, in all calculations pertaining to the DOS below, we also redefine the parameter $i\epsilon^{(1)} \to \epsilon^{(1)}$.

The second relevant observable — superfluid density $n_s$ — is calculated using the electromagnetic response kernel. The response kernel $Q_{\alpha\beta}(\mathbf{q}, i\omega_n)$ where $\omega_n = 2\pi T n$ (we use units with $k_B = 1$) determines the current in response to the applied electromagnetic field $A_\beta$,

$$j_\alpha(\mathbf{q}, i\omega_n) = Q_{\alpha\beta}(\mathbf{q}, i\omega_n) A_\beta(\mathbf{q}, i\omega_n), \quad (11)$$

where $\alpha$ and $\beta$ are two-dimensional spatial indices. Superfluid density is obtained as a zero frequency and zero momentum limit of the response kernel [47, 48]:

$$n_{s,\alpha\beta} = -\frac{m}{e^2} Q_{\alpha\beta}\left(\boldsymbol{q} = 0, i\omega_n = 0\right). \tag{12}$$

In the Appendix C we derive the expression for the response kernel using Green functions,

$$Q_{\alpha\beta}(0,0) = -\frac{e^2 T}{2} \sum_{\epsilon_n} \int_{\boldsymbol{k}} \left( \text{Tr}\left[\hat{v}_\alpha G\left(\boldsymbol{k}, i\epsilon_n\right) \hat{v}_\beta G\left(\boldsymbol{k}, i\epsilon_n\right)\right] - \text{Tr}\left[\hat{v}_\alpha G\left(\boldsymbol{k}, i\epsilon_n\right) \hat{v}_\beta G\left(\boldsymbol{k}, i\epsilon_n\right)\right]_{\Delta=0} \right), \tag{13}$$

where the velocity tensors $v_\alpha$ ($\alpha = x, y$) are defined as $\hat{v}_x = k_x/m$ and $\hat{v}_y = k_y/m$. Note, that here we neglected the contribution from spin-orbit coupling to velocity operator, since it is suppressed by the parameter $\lambda_{\text{so}}^2/v_F^2$, where $v_F$ is the Fermi velocity. Substituting the Eq. (13) into (12) and using the explicit form of the Green's function (7) we get that the tensor of superfluid densities is diagonal, with two non-zero components $n_{s,xx}$ and $n_{s,yy}$. The superfluid density $xx$-component is denoted as $n_{s,\perp}$ (since magnetic field points in $y$-direction) is given by the following expression

$$n_{s,\perp} = nT \sum_{\epsilon_n} \int d\xi \int \frac{d\phi_{\mathbf{k}}}{2\pi} \cos^2 \phi_{\mathbf{k}} \sum_{f=\pm 1} \frac{\left(\Delta^{(1)} + f\Delta^{(2)}\cos\phi_{\mathbf{k}}\right)^2 + \xi^2 - \left(\epsilon^{(1)} + if V^{(1)}\cos\phi_{\mathbf{k}}\right)^2}{\left[\left(\Delta^{(1)} + f\Delta^{(2)}\cos\phi_{\mathbf{k}}\right)^2 + \xi^2 + \left(\epsilon^{(1)} + if V^{(1)}\cos\phi_{\mathbf{k}}\right)^2\right]^2}. \tag{14}$$

The expression for the $yy$-component denoted as $n_{s,\parallel}$ can be obtained by replacing the $\cos^2\phi_{\mathbf{k}}$ with $\sin^2\phi_{\mathbf{k}}$.

We note, that the spin-orbit interaction strength $\lambda_{\text{so}}$ does not explicitly enter the expressions for superfluid density and DOS due to change of variables of integration. We use $\xi$ that is the relative energy difference to the respective spin-orbit split Fermi surfaces and neglect the difference in DOS between two Fermi surfaces assuming that spin-orbit splitting is much smaller compared to the Fermi energy. At the same time, presence of spin-orbit energy splitting that is much larger than induced pairing and Zeeman term (we assume that renormalization due to disorder does not change the order of magnitude of parameters $V^{(1)}, \Delta^{(1,2)}$) is important for our treatment, since this allows to neglect inter-band terms.

## 2.5 Arbitrary magnetic field direction and anisotropic $g$-tensor

Until now we considered the magnetic field pointing along $y$-direction and did not take into account possible $g$-tensor anisotropy. However, $g$-tensor anisotropy is generically expected in the presence of spin-orbit coupling. Indeed, in the relevant case of spin-orbit coupled asymmetric quantum wells, in-plane $g$-factor anisotropy is known to be a large effect [49–51].

Here we generalize our previous results to include both of these additional ingredients. Assuming general in-plane magnetic field and arbitrary $g$-tensor, $\hat{g} = \begin{pmatrix} g_{xx} & g_{xy} \\ g_{yx} & g_{yy} \end{pmatrix}$, the Zeeman contribution to energy is given by the convolution of Pauli matrices and magnetic field, $\mu_B \sigma^\alpha \hat{g}_{\alpha\beta} H_\beta/2$. Explicit convolution gives the expression $-\sigma^x V_x - \sigma^y V_y$ for the Zeeman energy that was previously given by $-\sigma^y V$ in Eq. (2). However, now the parameters $V_{x,y}$ are defined as:

$$V_x = \frac{1}{2}\mu_B H \left(g_{xx}\cos\chi + g_{xy}\sin\chi\right), \tag{15}$$

$$V_y = \frac{1}{2}\mu_B H \left(g_{yx}\cos\chi + g_{yy}\sin\chi\right), \tag{16}$$

where $\chi$ is the angle between magnetic field $\mathbf{H}$ and $x$-axis. Using these notations, we obtain the more general form of the Hamiltonian (2) in momentum representation,

$$h_0(\mathbf{k}) = \tau^z \xi_{\mathbf{k}} + \lambda_{so} k_F [\tau^z \sigma^y \cos \phi_{\mathbf{k}} - \sigma^x \sin \phi_{\mathbf{k}}] - \sigma^y V_y - \sigma^x V_x - \tau^y \sigma^y \Delta. \quad (17)$$

We can reduce this Hamiltonian to the situation considered previously by unitary transformation acting in the spin and Nambu spaces. To this end, we define the angle $\theta$ as

$$\sin \theta = -\frac{V_x}{V}, \quad \cos \theta = \frac{V_y}{V}, \quad V = \sqrt{V_x^2 + V_y^2}. \quad (18)$$

Rotating the Hamiltonian using unitary matrix $U_\theta$, $U_\theta = \cos \frac{\theta}{2} - \sin \frac{\theta}{2} \tau^z \sigma^z$ as $U_\theta^\dagger h_0(\mathbf{k}) U_\theta$ we bring it to the same form as before, Eq. (2), however with the shifted angle $\phi_{\mathbf{k}}$ and redefined $V$: $\phi_{\mathbf{k}} \to \phi_{\mathbf{k}} - \theta$,

$$V(H, \chi) = \frac{1}{2} g(\chi) \mu_B H, \quad (19)$$

where we introduced

$$g(\chi) = \sqrt{(g_{xx} \cos \chi + g_{xy} \sin \chi)^2 + (g_{yx} \cos \chi + g_{yy} \sin \chi)^2}. \quad (20)$$

Using this insight, we reexamine our calculation of physical quantities in Sec. 2.4. The angle rotation does not affect the calculation of DOS, since the shift of angle $\phi_{\mathbf{k}}$ does not change the integral over $\phi_{\mathbf{k}}$. Hence, the DOS in presence of $g$-factor anisotropy and general direction of magnetic field can be obtained from Eq. (10) using the generalized expression for $V$ from Eq. (19).

The extension of the calculation of the superfluid density is more involved, since the shift of the angle $\phi_{\mathbf{k}}$ affects Green's functions, but does not affect velocity operators in Eq. (13). Careful incorporation of such shift in Eq. (14) gives the following expression for the $xx$-component of the superfluid density that relies on the expressions $n_\perp(H)$ and $n_\parallel(H)$ introduced in Eq. (14),

$$n_{xx}(H, \chi) = \frac{(g_{yx} \cos \chi + g_{yy} \sin \chi)^2}{g^2(\chi)} n_\perp(V(H, \chi)) + \frac{(g_{xx} \cos \chi + g_{xy} \sin \chi)^2}{g^2(\chi)} n_\parallel(V(H, \chi)). \quad (21)$$

Note, that in the case of anisotropic $g$-tensor, the tensor $n_{\alpha\beta}$ is not diagonal anymore. Its remaining components can be calculated in the same way as $n_{xx}$ by incorporation of the shift of angle $\phi_{\mathbf{k}}$ in the Green function. In detail, for $xx$-component we performed the shift of the angle $\phi_{\mathbf{k}}$ in the Eq. (14) everywhere except the factor $\cos \phi_{\mathbf{k}}$ which corresponds to the velocity operators. For the $yy$ and $xy$-components one may use the same equation with $\cos^2 \phi_{\mathbf{k}}$ replaced by $\sin^2 \phi_{\mathbf{k}}$ and $\sin \phi_{\mathbf{k}} \cos \phi_{\mathbf{k}}$, respectively. This gives non-vanishing $n_{xy} = n_{yx}$ component of the superfluid density,

$$n_{xy}(H, \chi) = \left[ n_\parallel(V(H, \chi)) - n_\perp(V(H, \chi)) \right] \frac{(g_{xx} \cos \chi + g_{xy} \sin \chi)(g_{yx} \cos \chi + g_{yy} \sin \chi)}{g^2(\chi)}. \quad (22)$$

In particular, for two orthogonal orientations of magnetic field, $H \| x$ and $H \perp x$ we get

$$n_{xy}(H, 0) = \frac{g_{yx} g_{xx}}{g_{xx}^2 + g_{yx}^2} \delta n_{\parallel \perp}, \quad (23)$$

$$n_{xy}(H, \frac{\pi}{2}) = \frac{g_{xy} g_{yy}}{g_{yy}^2 + g_{xy}^2} \delta n_{\parallel \perp}, \quad (24)$$

where

$$\delta n_{\parallel, \perp} = n_\parallel \left( \frac{\mu_B H}{2} \sqrt{g_{xx}^2 + g_{yx}^2} \right) - n_\perp \left( \frac{\mu_B H}{2} \sqrt{g_{xx}^2 + g_{yx}^2} \right). \quad (25)$$

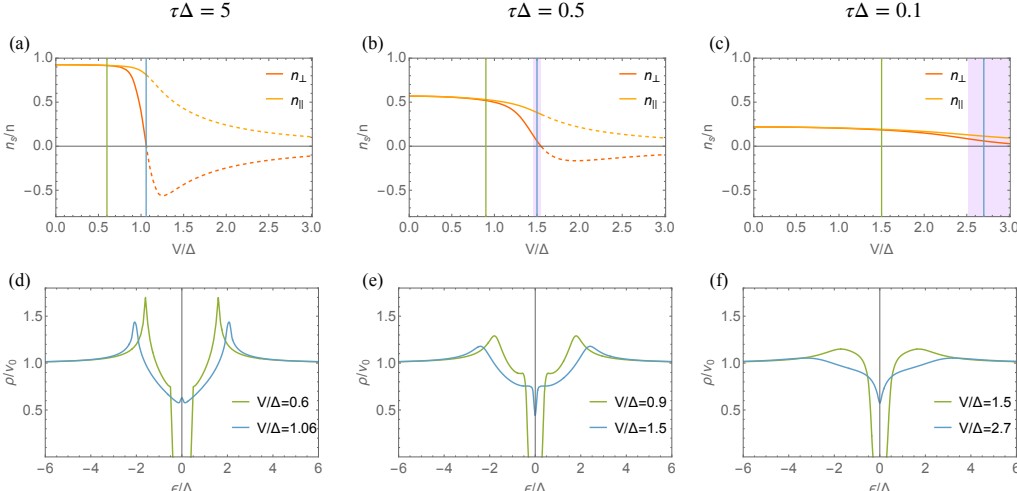

Figure 2: (a-c) Superfluid density normalized by the total electron density, $n_s/n$ as a function of magnetic field for three progressively increasing values of disorder. With increasing disorder the purple areas, representing the regions of magnetic field where there is a gapless regime, expand, and the decline of the superfluid density with magnetic field becomes smoother. In addition, the point where superfluid density becomes negative, so that our treatment is not applicable anymore (shown as dashed lines) is close to the clean case value $V = \Delta$ in panel (a) and is shifted to progressively larger values of magnetic field. (d-e) DOS normalized by its value in the normal state, $\rho/\nu_0$ for the two different values of magnetic field indicated by vertical lines in the upper panels. The larger value of the magnetic field puts the 2DEG into the gapless regime, as is clear from the non-zero value of the DOS at zero energy. For the calculation of superfluid density, the temperature was set to $T/\Delta = 0.0564$, and suppression of the order parameter by magnetic field is ignored in all plots.

From here we see that off-diagonal terms of $g$-tensor induce off-diagonal contributions to the superfluid density. Such non-diagonal $g$-tensors naturally arise in the anisotropic quantum wells of interest here [50]. In spin-3/2 hole systems within low-symmetry quantum wells (e.g., GaAs), the g-tensor can exhibit asymmetry due to the interaction between the p-like character of hole wave functions and asymmetric bandedge profiles, alongside specific crystallographic orientations [52]. Equations (23)-(24) suggest that measurements of the off-diagonal components of the superfluid density for different orientations of the magnetic field can be used to probe asymmetry between off-diagonal elements of the $g$-tensor. In particular, the extremely non-symmetric form of the $g$-tensor, with i.e. $g_{yx} \gg g_{xy}$ would lead to the signature $n_{xy}(H,0) \gg n_{xy}(H,\pi/2)$ potentially observable without quantitative fitting.

## 2.6 Review of depairing theory in conventional superconductor

One of the main approximations of the current theoretical model is the assumption that the order parameter $\Delta$ in the 2DEG is the same as in the superconductor. Thus although the main focus of our work is the theory of 2DEG, we need to know the superconducting order parameter at non-zero field and temperature. The in-plane magnetic field suppresses the order parameter in the superconductor according to the conventional depairing theory [40, 41] that is reviewed below for the sake of completeness.

To find the dependence of the superconductor order parameter on the magnetic field (still parametrized by $V$) and temperature, $\Delta(V, T)$ we assume that the superconducting layer is thick enough (unlike the semiconducting one) to neglect Zeeman contribution and leave only

the orbital contribution of an in-plane magnetic field [42]. At the same time we note that the thickness of the superconducting layer still has to be much smaller than the London penetration depth since we neglect the Meissner effect. In this case, the general approach of pair-breaking treatment in dirty superconductors is applicable [40–42]. This approach establishes the following connection between the critical temperature in absence of pair-breaking, $T_{c0}$, and the critical strength of pair-breaking $\alpha$ at fixed temperature $T$:

$$\ln\left(\frac{T}{T_{c0}}\right) + \psi\left(\frac{1}{2} + \frac{\alpha_c}{2\pi k_B T}\right) = \psi\left(\frac{1}{2}\right). \tag{26}$$

Here $\psi$ – is the digamma function, and the parameter $\alpha$ can incorporate contributions from various pair-breaking mechanisms, such as magnetic impurities, magnetic field, electric current, etc.

Using the Eq. (26) for a fixed temperature $T$, we can find the value $\alpha_c(T)$ – the value of pair-breaking parameter at which order parameter becomes equal to zero at temperature $T$. Recalling that in our case the physical nature of pair-breaking parameter $\alpha$ is magnetic field and that orbital contribution from in-plane magnetic field in thin film (the thickness of film is much smaller than coherence length) results in quadratic contribution [40]: $\alpha \propto V^2$, we can express the dependence of $\alpha$ on magnetic field in the following way:

$$\alpha = \alpha_c(T)\frac{V^2}{V_c^2}, \tag{27}$$

where $V_c$ is the critical magnetic field for the given temperature $T$ and is determined by intrinsic properties of specific superconductor. We notice that at zero temperature $\alpha_c(T = 0)$ has the following form: $\alpha_c(0) = \Delta_{00}/2 \equiv 2\pi k_B T_{c0} e^{\psi(1/2)}$, where $\Delta_{00}$ – order parameter at zero temperature and zero magnetic field, $T_{c0}$ - critical temperature at zero magnetic field and we used the well-known from BCS theory relation [53]. In the next Section we will consider a finite but very small temperature, $T = 0.0564\Delta_{00}$ so that $\alpha_c(T) = 0.49\Delta_{00}$ is very close to zero-temperature critical field, but at the same time small finite temperature provides a natural regularization for numerical calculations. Knowing the explicit dependence (27) of pair-breaking parameter $\alpha$ on magnetic field, we can find the order parameter $\Delta(T, V)$ from the following equations [41],

$$\sqrt{1 - f_n^2}\,(\Delta - f_n\alpha) = 2\pi T\,(n + 1/2)\,f_n\,, \tag{28}$$

$$\ln\left(\frac{T}{T_{c0}}\right) = \sum_{n=0}^{\infty}\left(\frac{2\pi T}{\Delta(T,V)}f_n - \frac{1}{n + 1/2}\right). \tag{29}$$

Here $f_n$ is the Eilenberger Green's function determined using numerical solution of Eq. (28) and substituted into Eq. (29) to obtain the self-consistent value of the gap at non-zero temperature and magnetic field.

Knowing the self-consistent gap value and Eilenberger Green's function $f_n$ also allows for calculation of the superfluid density as [1, 41]

$$\frac{n_s(T,V)}{n} = 2\pi T\tau_{\text{SC}}\sum_{n=0}^{\infty}\frac{f_n^2}{\frac{\tau_{\text{SC}}\Delta(T,V)}{f_n} + \frac{1}{2}}, \tag{30}$$

where $\tau_{\text{SC}}$ is a mean free path in superconductor that is assumed to satisfy $\tau_{\text{SC}}\Delta \ll 1$, and $n$ is the density of electrons in superconductor. The last equation for superfluid density $n_s$ of SC is not used in the next Section, because we analyze only superfluid density in the proximitized 2DEG. However, this expression will be used in the Section 4 where fit the experimental data, since experiment is probing the total superfluid density of 2DEG and SC.

# 3 Results for proximity-induced superconductivity in disordered 2DEG

In this section we study the influence of disorder on the proximity effect in the 2DEG using methods introduces in the previous section. First, we describe the numerical procedure used for the calculation of the self-consistent Green's function. Then we present our results for DOS and superfluid density. Finally, we focus on the region of gapless superconductivity and discuss the range of magnetic fields and disorder strengths where it occurs. We note that in the first two subsections we do not consider the suppression of order parameter $\Delta$ due to the magnetic field, and include it only in Sec. 3.3.

## 3.1 Numerical solution for Green's function

In order to find the renormalized parameters and DOS we numerically solve the system of self-consistent equations (8) using iterations: we substitute an iteration $n-1$ in the right side of an equation to obtain the value of the corresponding renormalized parameter at the next iteration, $n$, on the left side. For the calculation of DOS we need Green's function for real values of energies, which is why we make the analytic continuation in Eqs. (8) $i\epsilon_n \rightarrow \epsilon + i0$ and $i\epsilon^{(1)} \rightarrow \epsilon^{(1)}$ in the same way we did in Eq. (10). As an initialization of the iterative procedure, we use the following values of parameters: $\epsilon_{(0)}^{(1)} = \epsilon + i/(2\tau)$, $V_{(0)}^{(1)} = V$, $\Delta_{(0)}^{(1)} = \Delta$, and $\Delta_{(0)}^{(2)} = 0$ [imaginary part of $\epsilon_{(0)}^{(1)}$ is motivated by the fact that it reproduces DOS in normal state after substitution into Eq. (10)]. We use from 100 to 200 iterations in order to get convergent results, the stronger the disorder is, the more iterations were required. We note that iterative procedure can be performed independently at each energy separately. As a maximum value of the considered energy we use: $\epsilon_{\max} = 6\Delta$, so, $\epsilon \in [-\epsilon_{\max}, \epsilon_{\max}]$, since no additional feature were observed beyond this range. In addition, we discretize the values of $\epsilon$ and $V$ using the grid with the step of $\delta = 0.1\Delta$, and obtain functions at intermediate values of these parameters by interpolation.

For the calculation of superfluid density, we need to find Green function for imaginary frequencies $i\epsilon_n$ using the Eqs. (8) and the same iterative procedure. In that case, we use the following values of parameters to seed the iteration procedure: $i\epsilon_{(0)}^{(1)} = i\epsilon_n$, $\Delta_{(0)}^{(1)} = \Delta$, $V_{(0)}^{(1)} = V$ and $\Delta_{(0)}^{(2)} = 0$. For imaginary energies, numerical calculations converge faster, hence we use from 20 to 50 iterations. The temperature at which calculations are performed is $T = 0.0564\Delta$. Since temperature is finite, Matsubara energies are discrete, so, discretization is introduced only for the Zeeman energy $V$ with the step $\delta = 0.1\Delta$. The maximal value for the considered energy is $\epsilon_{\max} = 40 \cdot 2\pi T$.

## 3.2 DOS and superfluid density

Using the iteration procedure described above, we calculate the self-consistent Green's function across a range of disorder strengths. Figure 2 illustrates how the superfluid density is suppressed by magnetic field for three different values of disorder (zero-field values of superfluid density are in agreement with well-known analytical results [47]). At the same time, bottom panels of this figure illustrate the behavior of DOS for two typical values of magnetic field, the first being in the regular and second in the gapless regime (see discussion below).

At the smallest value of disorder, shown in Fig. 2(a) and (d), it has a weak effect on the superfluid density and behavior of DOS. Indeed, comparing Fig. 2(d) with Fig. 1(d) and (f) for the DOS in clean case we see qualitatively similar behavior. Moreover, the superfluid density $n_\perp$ in Fig. 2(a) shows a sharp decline to zero near the value of magnetic field that is only slightly larger then $V = \Delta$, predicted to give rise to Bogoliubov's Fermi surfaces in the clean case [6].

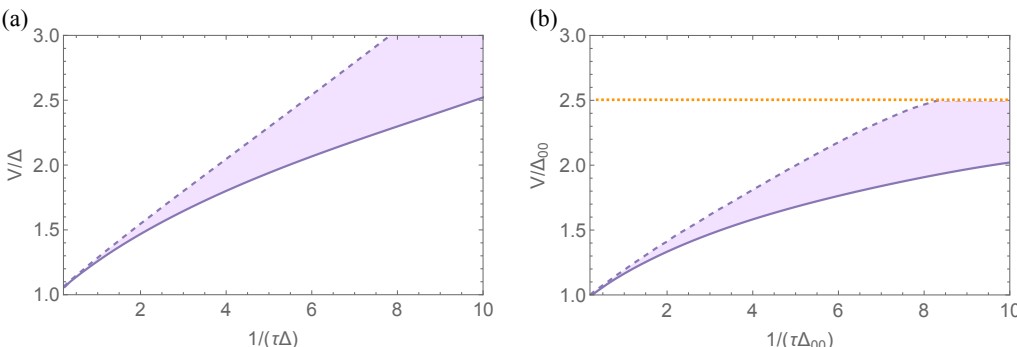

Figure 3: The span of magnetic fields where gapless superconductivity is realized (filled region between solid and dashed violet lines) expands with increasing disorder strength. The solid violet line corresponds to the onset of the gapless regime, dashed violet line marks the point when superfluid density turns negative, signaling that our model becomes inapplicable. Data shown in panel (a) assumes that the order parameter is not suppressed by magnetic field, panel (b) takes the depairing into account assuming critical magnetic field: $V_c = 2.5\Delta_{00}$, orange dotted line corresponds to this critical field $V_c$.

We note that nearly at the same time as the DOS becomes non-zero at $\epsilon = 0$, the $n_\perp$ component of the superfluid density turns negative. This makes the theory used here inapplicable, as is shown by dashed lines in Fig. 2(a). Thus, the *gapless regime* — situation where 2DEG has non-zero proximity-induced order parameter, but no gap exists in the density of states — may be realized only in the extremely narrow range of magnetic fields at weak disorder. This will be discussed in more details in Sec. 3.3 below.

When disorder strength is increased so that scattering time becomes twice shorter than inverse gap, its effect becomes more pronounced. Figure 2(b) and (e) reveal that even at zero magnetic field, $V = 0$ the superfluid density is already suppressed by about 50% compared to the total carrier density. In addition, the decline of the superfluid density becomes more gradual and peaks in the DOS at energies where clean system has Van Hove singularities become much smoother. Most importantly, the gapless superconductivity is now realized in a narrow window of magnetic field near the value of $V \approx 1.5\Delta$. Remarkably, the disorder not only "smears" all the singularities in the energy space, but also pushes the onset of rapid decline of superfluid density to significantly larger values of magnetic field and creates a gapless regime.

Finally, increasing disorder by additional factor of five in Fig. 2(c) and (f) results in a broad gapless regime that occurs for $V \gtrsim 2.5\Delta$ and extends beyond the maximal value of $V = 3\Delta$ considered in this work. DOS in Fig. 2(f) has extremely broadened peaks at the energy of Van Hove singularities in the clean case and develops a broad dip at zero energy upon entering the gapless regime. Remarkably, the onset of negative superfluid density is pushed beyond the considered range of magnetic fields. This may be understood as an effect of renormalization of effective magnetic field by disorder as we discuss in Appendix F.

The effect of disorder on the system is encoded in the self-consistent solution for the Green's function. In Appendix F we consider the results for renormalized singlet and triplet components of the order parameter, $\Delta^{(1)}$ and $\Delta^{(2)}$, that become now non-trivial functions of energy. In particular, we show that the disorder-induced triplet component has real part that is odd in frequency [54]. In addition, we also calculate the data for superfluid density and DOS including the order parameter suppression in superconductor in Appendix G. Upon comparing Fig. 10 from this appendix with the data in Fig. 2, we conclude that while the order parameter's suppression does not introduce new qualitative features, it does result in the expected vanishing of the superfluid density at the superconductor's critical field.

## 3.3 Region of the gapless superconductivity and potential instability

It is clear from the Fig. 2 that the extent of the gapless regime increases with disorder strength. In order to quantify this phenomenon, we numerically calculated DOS $\rho(\epsilon)$ at $\epsilon = 0$ for various values of disorder and magnetic field using the discrete grid with the step of $\delta = 0.02\Delta$ (or $\delta = 0.02\Delta_{00}$ when we incorporate the order parameter suppression) for magnetic field. For a specific value of disorder strength, with the increase of the magnetic field from 0, initially, $\rho(0)$ takes values much smaller than 1. At a certain (discrete) value of the magnetic field, $\rho(0)$ abruptly increases and becomes of the order of 1. We identify this value as the magnetic field at which a gapless regime emerges. For the numerical calculation we use the same approach discussed in Sec. 3.1. However, at the boundary between gapless and standard proximity induced superconducting regime more iteration — from 100 to 10000 — are required for convergence.

In Fig. 3(a) and (b) we present the area of parameters (in $1/\tau$ and $V$ axes) which corresponds to the gapless region. In Fig. 3(a) the suppression of order parameter by magnetic field is not taken into account while in Fig. 3(b) we apply the standard depairing theory reviewed in Sec. 2.6 for fixed low temperature $T = 0.0564\Delta_{00}$ and critical field of superconductor being $V_c = 2.5\Delta_{00}$, where $\Delta_{00}$ is the order parameter in superconductor at zero field and zero temperature. The solid purple line on the bottom of the region represents the values of $\tau$ and $V$ at which the superconducting gap disappears, the dashed line on the top of the region – values of $\tau$ and $V$ beyond which superfluid density becomes negative. As we discuss below in greater detail, the negative superfluid density signals that the current theoretical model is inapplicable. Nevertheless, the region of gapless superconductivity is vastly expanding with increasing disorder.

The prediction of the negative superfluid density may be viewed as a results of oversimplification that neglects inverse proximity effect while assuming perfectly transparent interface between 2DEG and superconductor. A more complete model should account for the inverse proximity effect, possibly leading to a reduction in critical magnetic field and change of order parameter suppression, thereby addressing the issue of negative superfluid density. To this end, incorporation of the free energy stemming from both 2DEG and superconductor is necessary. The self-consistent equations for the order parameter resulting from such total free energy would incorporate the inverse proximity effect. In addition, it would also allow to check if the state with spatially modulated order parameter of the form $\Delta(\mathbf{r}) = \Delta_s + \tilde{\Delta}e^{i\mathbf{q}\cdot\mathbf{r}}$ with $|\Delta_s| \gg |\tilde{\Delta}|$ is more favorable compared to the uniform state. We should specify, that the leading contribution to the order parameter cannot depend on coordinate because it comes from the superconducting layer which is strongly disordered, therefore, FFLO [55,56] phase is suppressed [18], besides, there is no spin-orbit coupling in the superconducting layer, thus, long-wave helical phase [18] also is not expected to appear. Meanwhile, the small correction comes from the 2DEG due to an inversed proximity effect and can depend on the spatial coordinate because the disorder is not assumed to be strong and, in addition, spin-orbit coupling is present. This model significantly differs from the model in Ref. [57] where the dependence of order parameter on coordinate was defined entirely by the 2DEG properties and as a result, there was no constant contribution from the superconducting layer.

The development of a more comprehensive theory discussed above is beyond the scope of this paper and could be an avenue for future research. At present, it is unclear if developing more realistic model will just quantitatively alter the predictions in vicinity of point where superfluid density turns negative, or if a non-uniform superconducting state may emerge above the dashed line in Fig. 3. Nevertheless, the prevalence of negative superfluid density diminishes with increasing disorder (as is seen on Fig. 3), making the current model applicable in a wider range of magnetic field. Moreover, despite the mentioned limitation, the current theory offers significant predictive power, as will be demonstrated in the next section by fitting experimental data.

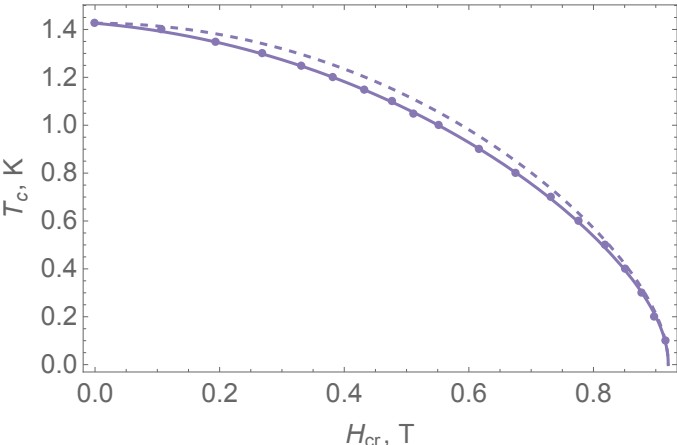

Figure 4: Fitting the theoretical model (solid line) to the experimental dependence (dots) of critical temperature on a magnetic field gives an excellent agreement for the value of fitting parameter $\zeta_1 = 0.207$. The dashed line corresponds to the fit in the absence of linear contribution from magnetic field to pair-braking parameter ($\zeta_1 = 0$).

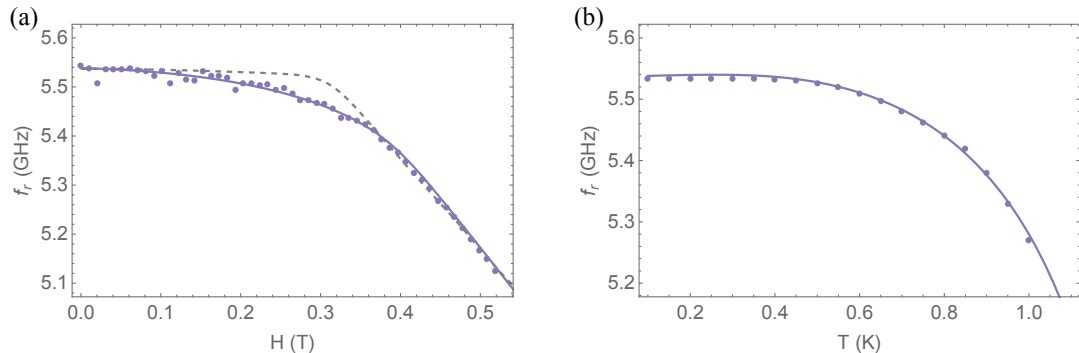

Figure 5: The theoretical model that incorporates disorder (solid lines) results in a much better agreement with the experimental data (dots), compared to the model [1] that does not incorporate non-magnetic disorder (dashed line). The data in panel (a) corresponds to the resonance frequency suppression with field $H_x$ at fixed $T = 0.1\,\mathrm{K}$. Panel (b) shows the temperature dependence of the resonant frequency at $H = 0\,\mathrm{T}$.

## 4 Fitting experimental data

In this section, we apply the theoretical model described above to the experimental data on Al-InAs heterostructure reported in Ref. [1]. Since the experiment probes the total superfluid density, first in Sec. 4.1 we fit the dependence of pair breaking parameter in aluminum. Next, in Sec. 4.2 we discuss the fitting procedure that we use to fit the experimental data and extract material parameters. Finally, in Sec. 4.3 we discuss the extracted values of parameters and show predictions for the DOS that can be checked in future experiments.

### 4.1 Pair-breaking effects in Al

In order to understand the dependence of the order parameter $\Delta$ in Al on temperature $T$ and magnetic field $H$ we apply the depairing theory reviewed in Sec. 2.6. Specifically, we use the dependence of the critical temperature on magnetic field measured in Ref. [1] to determine

the pair-breaking parameter $\alpha$ as a function of magnetic field. In contrast to Ref. [1] that extracted the depairing parameter as a phenomenological function of magnetic field, here we aim to capture the dependence of depairing parameter by a simple function thereby providing insights into depairing mechanisms in aluminum. Moreover, we emphasize that although experimental data from Ref. [1] measures the critical temperature of the 2DEG-superconductor heterostructure, we ignore the effect of the 2DEG onto superconductor. This is consistent with the approximation of neglecting the inverse proximity effect adopted in this work.

In Sec. 2.6 we discussed that the orbital contribution from in-plane magnetic field results in the quadratic dependence $\alpha \propto H^2$. However, assuming purely quadratic dependence of depairing on the field does not result in a good quantitative agreement with the experimental data (see dashed line in the Fig. 4). This discrepancy may be consistent with the tiny out-of-plane component of magnetic field that, despite careful alignment may be present in the experiment, see Appendix E. Alternatively it may potentially emerge from the influence of magnetic field on the inverse proximity effect that is not considered in the present work. The out of plane component of magnetic field contributes linearly to the depairing parameter $\alpha$ [42, 58, 59]. Thus we use the following ansatz for the depairing parameter

$$\alpha(H) = 2\pi T_{c0} e^{\psi(1/2)} \left( \zeta_1 \frac{H}{H_{\text{cr}}} + \zeta_2 \frac{H^2}{H_{\text{cr}}^2} \right), \tag{31}$$

where two dimensionless constants $\zeta_{1,2}$ parametrize the entire dependence. Here, $T_{c0}$ is the critical temperature in the absence of the field, and $H_{\text{cr}}$ is the zero-temperature critical magnetic field.

We can establish additional relation between constants $\zeta_{1,2}$ introduced above. For this we consider the vicinity of critical field $H \to H_{\text{cr}}$. For such values of magnetic field, the critical temperature tends to zero $T_c \to 0$ and we can simplify the relation (26) from Sec. 2.6 by expanding it assuming that $T_c \to 0$. This expansion gives us the following relation

$$\alpha(H_{\text{cr}}) = 2\pi T_{c0} e^{\psi(1/2)} \equiv \Delta_{00}/2. \tag{32}$$

Comparing this with Eq. (31) we derive relation between $\zeta_1$ and $\zeta_2$: $\zeta_1 + \zeta_2 = 1$. Therefore, we set $\zeta_2 = 1 - \zeta_1$ and keep $\zeta_1$ as the only fitting parameter, that intuitively encodes the relative weight of the linear-in-field contribution to the depairing parameter.

Using specific dependence of $\alpha$ on magnetic field in Eq. (31), we calculate the dependence of $T_c$ on applied magnetic field according to Eq. (26). The resulting curve is fitted to experimental data on the dependence of critical temperature on magnetic field using the least square method to determine the best value of $\zeta_1$. The fit has excellent agreement with experimental data, as shown in Fig. 4, yielding the following value of $\zeta_1$,

$$\zeta_1 = 0.207 \pm 0.002. \tag{33}$$

Knowledge of the explicit dependence of pair-breaking parameter $\alpha$ on magnetic field $H$ allows us to find the order parameter $\Delta(T, H)$ and superfluid density in Al using Eq. (28)-(30). Note, that we take $\tau_{\text{SC}} \Delta_{00} = 0.001$ for Al, which corresponds to the dirty limit, but the functional form of all dependences is not affected by specific value of $\tau_{\text{SC}}$ as soon as it is sufficiently small.

## 4.2 Fitting procedure

After determining the depairing in Al, we turn to fitting the data on the superfluid density measured by the experiment [1]. The experiment probes the Al-InAs heterostructure by placing it in resonator and sensitively measuring the resonance frequency $f_r$. This frequency has a constant "geometric" contribution denoted as $f_{\text{geo}}$, and also receives a contribution from

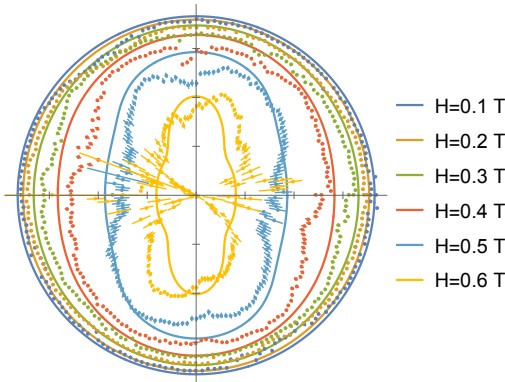

Figure 6: Polar plot for the dependence of resonance frequency on the angle between a magnetic field and $x$-axis, radial divisions start at $f_r = 4.8$ GHz and are in 0.2 GHz increments. Different colors encode the values of the magnetic field, with dots corresponding to experimental data and solid lines showing the best fit of the theoretical model.

superconducting condensate in aluminum and 2DEG, denoted as $f_{\text{kin}}$. The total resonance frequency measured in experiment reads [1]:

$$\frac{1}{f_r^2} = \frac{1}{f_{\text{geo}}^2} + \frac{1}{f_{\text{kin}}^2} . \tag{34}$$

Assuming the distributed circuit model, Ref. [1] suggested that the kinetic contribution to inductance may be viewed as a sum of 2DEG and superconductor contributions,

$$f_{\text{kin}}^2 = c_p \frac{n_s^{(\text{InAs})}(H, T)}{n^{(\text{InAs})}} + c_s \frac{n_s^{(\text{Al})}(H, T)}{n_s^{(\text{Al})}(0, 0)} . \tag{35}$$

Here we wrote a combination of two individual contributions from 2DEG and superconductor in a slightly different form, and emphasized the magnetic field and temperature dependence. Both contributions depend on the constants $c_{p,s}$ that have dimension of squared frequency and will be our fitting parameters. For the 2DEG the constant $c_p$ multiplies the dimensionless ratio between the superfluid density in 2DEG and normal carrier density. In contrast, for superconductor, the constant $c_s$ multiplies the superfluid density at a given field and temperature, normalized by the superfluid density at zero temperature and field, $n_s^{(\text{Al})}(0, 0) = \pi(\tau_{SC}\Delta)n^{(\text{Al})}$, that is suppressed by a small factor $\tau_{SC}\Delta$ compared to the full carrier density $n^{(\text{Al})}$ in the dirty limit. In both cases, the normalized superfluid densities will be provided by the theoretical model developed above.

After we obtain the value of parameters $c_p$ and $c_s$ from fitting, we may use them to extract material parameters. To this end we use on the following relation between these constants and material parameters, developed in Ref. [1]:

$$c_p = \frac{1}{4l^2 C\gamma} \frac{n^{(\text{InAs})}e^2}{m^{(\text{InAs})}} , \qquad c_s = \frac{1}{4l^2 C\gamma} \frac{\pi\Delta_{00}}{R_\square^{(\text{Al})}} . \tag{36}$$

Here $l$ denotes the resonator length, $C$ is the resonator capacitance, $\gamma$ is a geometric factor characterizing the resonator and given in Eq. (S4) in the supplement of Ref. [1], $m^{(\text{InAs})}$ is the effective mass of a quasiparticle in InAs, $e$ is the electron charge, $\Delta_{00}$ is zero-field zero-temperature order parameter in Al, and $R_\square^{(\text{Al})}$ is the aluminum sheet resistance in normal state. From here it is clear that knowledge of constants $c_{p,s}$ will yield an estimate of the carrier

Table 1: Result of the fitting procedure. Magnetic anisotropy is significant since $g_{xx}^{\text{eff}}$ differs from $g_{yy}^{\text{eff}}$. The disorder is intermediate meaning that dirty-limit and clean-limit approximations are not applicable to the specific Al-InAs heterostructure.

| Parameter | Fit value | Error |
|:---:|:---:|:---:|
| $c_p$ | 286 GHz$^2$ | 27 GHz$^2$ |
| $c_s$ | 72 GHz$^2$ | 4 GHz$^2$ |
| $f_{geo}$ | 6.08 GHz | 0.01 GHz |
| $g_{xx}^{\text{eff}}$ | 30.8 | 0.8 |
| $g_{yy}^{\text{eff}}$ | 17.8 | 0.8 |
| $\tau\Delta_{00}$ | 0.23 | 0.05 |

density in 2DEG and sheet resistance of superconductor in the normal state, provided we rely on independent estimates of remaining parameters in Eq. (36).

The fitting procedure used to match experimental data relies on Eqs. (34)-(35). These equations explicitly contain three fit parameters, $c_p$, $c_s$, and $f_{\text{geo}}$. The theoretical model for the ratio $n_s^{(\text{Al})}(H,T)/n_s^{(\text{Al})}(0,0)$ is entirely fixed by Eqs. (31) and (33) in previous section and contains no additional free parameters. In contrast, the normalized superfluid density of 2DEG additionally depends on the scattering time $\tau$ intrinsic to 2DEG and components of $g$-tensor, $g_{xx}^{\text{eff}}$ and $g_{yy}^{\text{eff}}$ (remaining components are assumed to be zero). Thus in total the fit between theoretical model and experimental data is performed over six parameters $c_p$, $c_s$, $f_{\text{geo}}$, $\tau$, $g_{xx}^{\text{eff}}$, and $g_{yy}^{\text{eff}}$. Note, that we denote the $g$-factors in the fitting as effective ones, since in what follows we phenomenologically incorporate the orbital effect to estimate their physical values.

In practice, to achieve such six-parameter fit we simultaneously fit three experimental dependencies as illustrated in Fig. 5 and Fig. 6. the dependence of $f_r$ on $T$ at $H = 0$ T, the dependence of $f_r$ on $H$ at $T = 0.1$ K and the dependence of $f_r$ on the angle between magnetic field and $x$-axis for six different values of magnetic field listed in Fig. 6. The simultaneous fit was conducted in the following manner: we constructed the cost function for each observable $F_\alpha$ as $S_\alpha = \sum_i \left( F_\alpha^{(\text{th})}(x_i) - F_\alpha^{(\text{exp})}(x_i) \right)^2$, where $F_\alpha^{(\text{th})}(x)$ is theoretical prediction for the observable $F_\alpha$ and $\{F_\alpha^{(\text{exp})}(x), x\}$ represents the experimental data. Specifically, we encode three experimental datasets as $\{F_1^{(\text{exp})}(x), x\} \equiv \{f_r(H), H\}$ for low-temperature magnetic field sweep shown in Fig. 5(a), $\{F_2^{(\text{exp})}(x), x\} \equiv \{f_r(T), T\}$ for zero-field temperature sweep in Fig. 5(b), and $\{F_3^{(\text{exp})}(x), x\} \equiv \{f_r(H,\chi), (H,\chi)\}$ for magnetic field and its direction sweep in Fig. 6. Then, to derive the total cost function, we sum these functions with specific weights $w_\alpha$. The weights are selected in such a way that at the minimum of the total cost function, contributions from different cost functions were approximately of the same order: $S_{\text{tot}}(c_p, c_s, f_{\text{geo}}, g_{xx}, g_{yy}, \tau) = w_1 S_1 + w_2 S_2 + w_3 S_3$. By minimizing this cost function, we extracted the best values of fitting parameters.

To estimate errors for these fitting parameters, we use a procedure similar to bootstrapping [60]. Specifically, we construct a set of cost functions $\{S_{\text{tot}}^{(j)}\}$ in the following way: for each observable, we select only a portion of the experimental points in a random manner $\{F_\alpha^{(\text{exp})}(x), x\}^{(j)}$ and then proceeded with the fitting in the same way as described above. This yields a set of different values for the fitting parameters $\{c_p, c_s, f_{\text{geo}}, g_{xx}^{\text{eff}}, g_{yy}^{\text{eff}}, \tau\}^{(j)}$. We check the resulting distribution of fit parameters and estimate error bars from its variance.

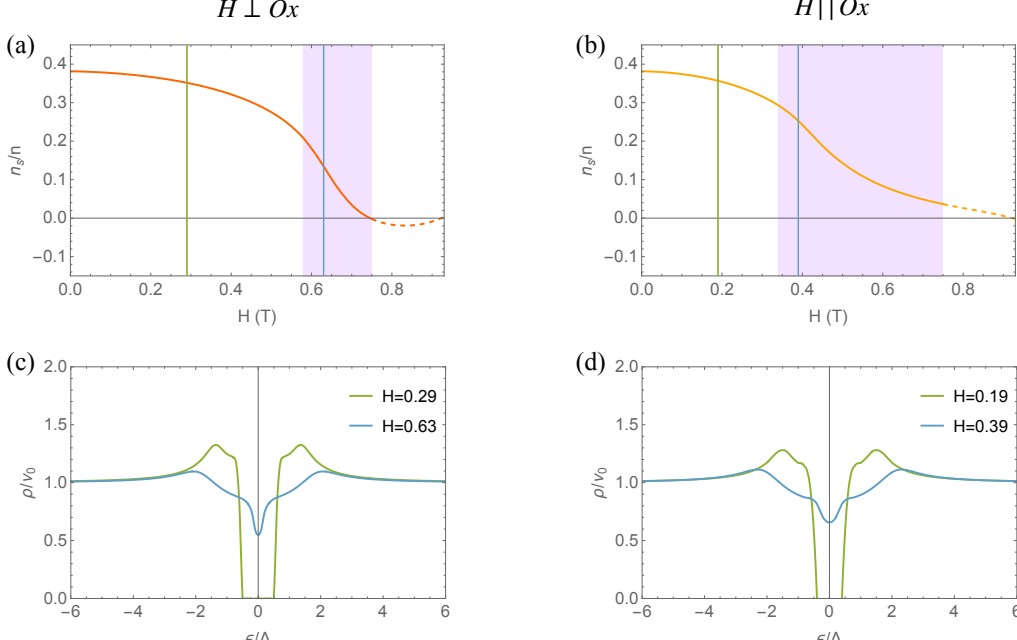

Figure 7: The material parameters extracted from the best fits to experimental data are used to generate predictions for superfluid density of 2DEG and DOS for two different orientations of magnetic field. Purple areas correspond to the regions of the magnetic field where the gapless regime is realized. Vertical lines on the plots for superfluid densities in (a)-(b) show the values of the magnetic field at which DOS is plotted on the respective panels (c)-(d), with green line illustrating the regime with a gap, while the blue line corresponding to the field when system is in the gapless regime.

### 4.3 Material parameters and prediction for DOS and gapless regime

The fitting of the theoretical model to experiment using the procedure described above gives the value of parameters that are summarized in Table 1. Figures 5-6 demonstrate the good quantitative agreement of best fits from the theory model with the experimental data. In particular, the agreement is much better compared to the model where disorder is not incorporated from Ref. [1], compare dashed and solid lines in Fig. 5. Also, due to presence of 1/2 in the definition of $V$, Eq. (3), the values of $g$-factor are now approximately two times larger.

Out of six fit parameters in Table 1, $g$-factors have the most immediate physical interpretation. However, the $g$-factors values of $g_{xx}^{\text{eff}} = 30.8 \pm 0.8$ and $g_{yy}^{\text{eff}} = 17.8 \pm 0.8$ are considerably larger compared to the ones used in the literature, where a range of values 3-11 was reported [61–64]. We note, that comparable values of $g$-factor can be obtained from the qualitative model in Ref. [1] that ignores disorder, provided one uses definition Eq. (3) for Zeeman energy. Such overestimation of $g$-factor points out that the orbital effect of the in-plane field cannot be neglected. For instance, recent work [43] reported similar overestimation of $g$-factor due to ignorance of orbital effects in a Josephson junction made out of similar superconductor-2DEG heterostructures.

While rigorous incorporation of the orbital effect of the magnetic field is reserved for the future work, we take this effect into account phenomenologically. Specifically, we assume that 2DEG is separated by a distance $d_{\text{2DEG}}$ from the superconducting layer. Then taking into account the phase accumulated to in-plane magnetic field results in the following induced order parameter in InAs, $\Delta_{\text{InAs}}(\mathbf{r}) = \Delta_{\text{Al}} \cdot e^{i2\phi(\mathbf{r})}$, where $\phi(\mathbf{r}) = \mathbf{q} \cdot \mathbf{r}$, with $|\mathbf{q}| = \pi H d_{\text{InAs}}/\Phi_0$,

where $\Phi_0$ is the magnetic flux quantum [57]. Proceeding in the same way as in Ref. [57], we derive that effect of such non-zero momentum $\boldsymbol{q}$ can be incorporated by changing the value of the parameter $V$ into $V^{(\text{eff})}$ with

$$V^{(\text{eff})} = V \pm v_F q = \left( \frac{1}{2} g \mu_B \pm \frac{\pi d_{\text{InAs}}}{\Phi_0} v_F \right) H, \tag{37}$$

where $\pm$ sign corresponds to different Rashba-split bands and we neglect the contribution $\lambda_{\text{so}} q$ since $\lambda_{\text{so}} \ll v_F$ is assumed throughout this work. We can refine our estimate of the $g$-factor by using the right hand side of Eq. (37). Equating expression in parenthesis to the fitted value of $g$-factor, we can extract its physical value according to expression:

$$g = g^{\text{eff}} - \frac{2\pi d_{\text{InAs}} v_F}{\mu_B \Phi_0}. \tag{38}$$

The extraction of $g$-factor in such phenomenological way strongly depends on the distance between 2DEG and superconductor. For instance, assuming distance $d_{\text{2DEG}} \sim 0.2\,\text{nm}$ we obtain 40% smaller value of $g_{xx} \sim 18$ and even more suppressed value of $g_{yy} \sim 5$. Larger distance of $d_{\text{2DEG}} \sim 0.5\,\text{nm}$ would result in a nearly vanishing value of $g_{xx}$. While these values of thickness are approximately an order of magnitude smaller compared to the estimated distance between aluminum and 2DEG layer, the band bending can decrease the effective distance. Also, our phenomenological considerations do not treat the orbital effect self-consistently, thus overestimating its magnitude. Although orbital effect for such thickness is (within our phenomenology) nearly equivalent to the Zeeman energy contribution, it does not involve the electron spin and hence is expected to be isotropic. Thus, we expect that quantitative incorporation of the orbital effect may allow for extraction of effective thickness as well as $g$-factors using the fitting of the data rather than phenomenological considerations.

We note that treatment discussed above is phenomenological, and proper incorporation of such orbital effect and disorder requires revisiting our self-consistent treatment presented above. In particular, we expect the renormalization due to disorder to suppress the orbital effect. Additional subtlety is related to the sign of the $g$-factor that is know to be negative in InAs, and to the relative $\pm$ sign in Eq. (37). Since our treatment is not sensitive to the relative sign of the $g$-factor, in order to obtain Eq. (38), we assume $g > 0$ and also choose the plus sign in Eq. (37) corresponding to the Fermi surface where gap closes earlier.

Next, we discuss the scattering time in 2DEG that is also extracted from the fitting. The value $\tau \Delta_{00} = 0.23$ suggests that InAs heterostructure is in intermediate between clean and dirty regimes, with its mean free path being of the order of $1.3\,\mu m$, assuming $\lambda_F = 10\,\text{nm}$. Estimating mobility from the scattering time we obtain

$$\mu = \frac{\tau e}{m^{(\text{InAs})}} = (3.1 \pm 0.7) \cdot 10^4 \, \frac{\text{cm}^2}{\text{V} \cdot \text{s}}, \tag{39}$$

where for the effective mass of electron in InAs we used $m^{(\text{InAs})} = 0.04 m_e$ [64], $m_e$ is electron mass, and $e$ is the charge of electron. The InAs mobility, measured in the absence of Al, is $1.3 \cdot 10^4 \, \text{cm}^2/(\text{V} \cdot \text{s})$ [1], suggesting that the presence of Al does not drastically alter the InAs carrier mobility.

Finally, the parameters $c_s$, $c_p$, and $f_{\text{geo}}$ provide consistency check. Parameter $f_{\text{geo}}$ is consistent with the expected value based on electromagnetic simulations in the Ref. [1]. Knowing the parameters $c_s$ and $c_p$ we can calculate the aluminum sheet resistance $R_{\square}^{(\text{Al})}$ and carrier density in InAs, $n^{(\text{InAs})}$, using Eq. (36):

$$R_{\square}^{(\text{Al})} = (7.3 \pm 0.4) \, \Omega, \tag{40}$$

$$n^{(\text{InAs})} = (8.0 \pm 0.8) \cdot 10^{13} \, \text{cm}^{-2}. \tag{41}$$

The resulting value of the aluminum sheet resistance is approximately the same as in Ref. [1]. However, the InAs carrier density is twice as large. The different estimate arises due to suppression of superfluid density by disorder that was not incorporated in the theoretical model used in Ref. [1]. This omission resulted in a lower value of $c_p$ compared to ours, which in turn led to lower values of the carrier density. Comparing the fit carrier density with the Hall value $n^{(\text{InAs})} = 1.06 \cdot 10^{12}\,\text{cm}^{-2}$, measured in the absence of Al, we find that the InAs carrier density is dramatically increased by the presence of Al.

Using the material parameters from the Table 1 we generate predictions for the DOS and superfluid density in proximitized InAs and check whether gapless regime is accessible. Fig. 7 shows our predictions for two orientations of magnetic field. When magnetic field is parallel to the $x$-axis we see that the system is predicted to have a wide gapless regime. In contrast, for the field aligned with $y$-axis we see the sharper decline of the superfluid density, gapless regime occurs at larger values of magnetic field, and has smaller extent. This is explained by the anisotropic response of superfluid density in 2DEG due to presence of spin orbit coupling. In principle, the dependence of DOS on energy can be studied experimentally using a scanning tunneling microscope. Thus, the predictions presented in Fig. 7 can be experimentally verified.

Finally, let us discuss the uncertainties in determining the parameters. We note, that although error bars estimated by bootstrapping in Table 1 are relatively small, the systematic uncertainties that come from potential further simplifications present in the model, such as phenomenological treatment of orbital effect, assumption of perfectly isotropic mass, consideration of only Rashba spin orbit coupling, diagonal form of $g$-tensor, and assumption of fully transparent interface between 2DEG and superconductor are potentially much larger. While relaxing these assumptions is straightforward, additional data is needed to prevent overfitting and enable reliable extraction of additional material parameters.

# 5 Summary and Outlook

To conclude, we examined a heterostructure consisting of 2DEG with strong spin-orbit coupling proximitized by superconductor and subjected to an in-plane magnetic field. Our theoretical model incorporates non-magnetic disorder of arbitrary strength in the semiconducting layer, a feature that was not addressed in previous studies. Our model gives predictions for the density of states and the superfluid density as a function of the magnetic field and varying disorder strength. We observe that increasing disorder stabilizes a gapless superconducting phase within a progressively increasing range of magnetic fields. This regime may be viewed as an extension of the phase with Bogoliubov Fermi surfaces that incorporates disorder in the system.

We applied our theoretical model to experimental data for an Al-InAs heterostructure obtained in Ref. [1]. Our model that incorporates disorder was able to quantitatively describe the data, also enabling us to extract parameters of the InAs layer, such as the anisotropic $g$-tensor, scattering time due to disorder, and mobility. Using extracted parameters of the heterostructure, we identified the range of magnetic fields, where the gapless regime of superconductivity is realized and generated predictions for the density of states that may be tested in future experiments.

Although our model was able to describe the experimental data and generate predictions, a number of questions remains open. First, while our theoretical model incorporates disorder compared to earlier theoretical studies, it still relies on a number of approximations. In particular, it may be desirable to relax the assumption of the perfectly transparent interface between 2DEG and superconductor, incorporate orbital effect of the magnetic field and inverse proximity effect – phenomena related to more realistic description of the motion of electrons between

2DEG and superconductor.[1] Incorporation of orbital effects may be particularly important for Germanium that has small value of $g$-factor and also for surface states of topological insulator proximitized by the bulk superconductor [25] where orbital contribution dominates the physics. In addition, to make the model of 2DEG more realistic, one may incorporate the Dresselhaus spin-orbit coupling along with Rashba spin-orbit considered here, and take into account more realistic band structure of 2DEG. These ingredients may be particularly important for treatment of heterostructures beyond Al-InAs, such as Al-Ge [39], Al-InSbAs [65], and other materials.

Bringing additional ingredients into our theoretical model, that already relied on a six-parameter fitting for describing experimental data, would most likely require additional experimental probes. In particular, it is easy to relax the assumption that the induced order parameter in the 2DEG is equal to its value in the superconductor, and relying on the tunneling measurements use more accurate relation. Using the framework for Green's function calculation set up in this work, one can easily calculate other experimental observables such as spin susceptibility and finite-frequency electromagnetic response kernel. Including more observables measured in situ, such as spin susceptibility [66], optical conductivity [48], density of states, or noise spectra [67] would enable even more detailed material characterization and allow for independent verification of our model.

Another set of questions that remain open is the eventual fate of the gapless proximity-induced superconducting state in 2DEG at sufficiently weak disorder and in the clean limit [1]. While the presence of superconducting film suggests that this instability will not destroy pairing in the system, it may cause reconstruction of the low energy band structure leading to reduced density of states. Understanding the instability requires incorporation of the inverse proximity effect and self-consistent treatment of the superconductor that takes into account the 2DEG, that are beyond the present work. Construction of such model may assist the future experiments in the identification and characterization of the phase diagram.

Finally, the experimental control available in the microwave cavity setups akin to Refs. [1] call for the theoretical development of nonequilibrium probes of 2DEG-superconductor heterostructures. In particular, the cavity enables to excite the system in the regime beyond liner response, and potentially probe it in the time-resolved fashion. In addition, electric contacts may enable to use current as an alternative pair-breaking mechanism to the in-plane magnetic field. These capabilities invite the theory of such 2DEG-superconducting system subject to nonequilibrium effects and currents, that would allow to gain further insights into the material properties and potentially uncover new phases.

## Acknowledgments

We acknowledge useful discussions with M. Geier, A. Levchenko, B. Ramshaw, T. Scaffidi, and J. Shabani.

**Funding information** This research was funded by the Austrian Science Fund (FWF) F 86. For the purpose of open access, authors have applied a CC BY public copyright licence to any Author Accepted Manuscript version arising from this submission. MS acknowledges hospitality of KITP supported in part by the National Science Foundation under Grants No. NSF PHY-1748958 and PHY-2309135. APH acknowledges the support of the NOMIS foundation.

---

[1]Note, that we assume the homogeneous penetration of magnetic field into superconducting film, implying the absence of vortices. This is a crucial ingredient for the applicability of our model, and it may be verified in the control experiment where the response of superconducting film without 2DEG is compared to standard depairing theory.

# A DOS in the clean case

In this Appendix we derive an analytic expression for the DOS of semiconductor in the absence of disorder. We use expression of Hamiltonian $h_0(\mathbf{k})$ in momentum representation, Eqs. (1) and (2) and make rotation in spin and particle-hole spaces, $\tilde{h}_0(\mathbf{k}) = U^\dagger h_0(\mathbf{k}) U$ with the following unitary matrix:

$$U = \frac{1}{\sqrt{2}} \begin{pmatrix} 1 & 0 & 1 & 0 \\ ie^{i\phi_\mathbf{k}} & 0 & -ie^{i\phi_\mathbf{k}} & 0 \\ 0 & 1 & 0 & 1 \\ 0 & ie^{-i\phi_\mathbf{k}} & 0 & -ie^{-i\phi_\mathbf{k}} \end{pmatrix}. \tag{A.1}$$

This rotation results in the following simplified form of the Hamiltonian,

$$\tilde{H}_0(\mathbf{k})$$
$$= \begin{pmatrix} \xi_\mathbf{k} + \lambda_{so}k_F - V\cos\phi_\mathbf{k} & i\Delta e^{-i\phi_\mathbf{k}} & iV\sin\phi_\mathbf{k} & 0 \\ -i\Delta e^{i\phi_\mathbf{k}} & -\xi_\mathbf{k} - \lambda_{so}k_F - V\cos\phi_\mathbf{k} & 0 & -iV\sin\phi_\mathbf{k} \\ -iV\sin\phi_\mathbf{k} & 0 & \xi_\mathbf{k} - \lambda_{so}k_F + V\cos\phi_\mathbf{k} & -i\Delta e^{-i\phi_\mathbf{k}} \\ 0 & iV\sin\phi_\mathbf{k} & i\Delta e^{i\phi_\mathbf{k}} & -\xi_\mathbf{k} + \lambda_{so}k_F + V\cos\phi_\mathbf{k} \end{pmatrix}. \tag{A.2}$$

Using the fact that $\lambda_{so}k_F \gg V$ we can neglect top right and lower left $2 \times 2$ blocks that correspond to interband coupling that connects different Rashba-split bands. This is equivalent to neglecting terms of the order $V^2/(\lambda_{so}k_F)^2$, and is justified for strong spin-orbit coupling. This allows us to calculate the Green function by inverting individual two by two matrix blocks, resulting in following two contributions labeled by $f = \pm 1$,

$$\tilde{G}(\mathbf{k}) = \sum_{f=\pm 1} \left(\frac{1 + f\sigma^z}{2}\right) \frac{\tau^z(\xi_\mathbf{k} + fk_F\lambda_{so}) + f\Delta(\tau^x\sin\phi_\mathbf{k} - \tau^y\cos\phi_\mathbf{k}) + (\omega + fV\cos\phi_\mathbf{k})}{-\Delta^2 - (\xi_\mathbf{k} + fk_F\lambda_{so})^2 + (\omega + fV\cos\phi_\mathbf{k})^2}. \tag{A.3}$$

Substituting this into expression for DOS results in the following integral,

$$\rho(\omega) = \frac{\nu_0}{2\pi^2} \text{Im} \int_0^{2\pi} d\phi \int_{-\infty}^{\infty} d\xi \frac{V\cos\phi + \omega}{\xi^2 + \Delta^2 - (V\cos\phi + \omega + i0)^2}. \tag{A.4}$$

In order to calculate this integral we proceed as follows: first, we introduce complex variable $z = e^{i\phi}$ and transform the integral over $\phi$ into the integral of complex variables over the cycle $|z| = 1$. Then, making a change of variables $\xi = \Delta\sinh\text{arccosh}x$ and using formula 3.148.6 from Ref. [68] we integrate over $x$. As a result, after introducing the following short-hand notation,

$$\Omega_{\mu\eta} = 1 + \mu\frac{\omega + \eta V}{\Delta}, \quad \mu, \eta = \pm, \tag{A.5}$$

for $V < \Delta$ we obtain,

$$\rho(\omega) = \frac{\nu_0}{2\pi} \begin{cases} 0, & 0 \le \omega < \Delta - V, \\ \frac{2}{\sqrt{V/\Delta}}\left[\Omega_{--}\Pi\left(\frac{-\Omega_{-+}}{2V/\Delta}, \frac{-\Omega_{-+}\Omega_{+-}}{4V/\Delta}\right) + (1 - \Omega_{--})K\left(\frac{-\Omega_{-+}\Omega_{+-}}{4V/\Delta}\right)\right], & \Delta - V < \omega < \Delta + V, \\ \frac{4}{\sqrt{-\Omega_{-+}\Omega_{+-}}}\left[K\left(\frac{4V/\Delta}{-\Omega_{-+}\Omega_{+-}}\right) - \Omega_{--}\Pi\left(\frac{2V/\Delta}{-\Omega_{-+}}, \frac{4V/\Delta}{-\Omega_{-+}\Omega_{+-}}\right)\right], & \Delta + V < \omega. \end{cases} \tag{A.6}$$

The DOS is even function, $\rho(\omega) = \rho(-\omega)$ so we defined it only for positive values of energy. For $V > \Delta$ we have,

$$
\rho = \frac{\nu_0}{2\pi}
\begin{cases}
\frac{4}{\sqrt{\Omega_{--}\Omega_{++}}}\left[2\Pi\left(\frac{-\Omega_{+-}}{\Omega_{--}}, \frac{\Omega_{+-}\Omega_{-+}}{\Omega_{--}\Omega_{++}}\right) - K\left(\frac{\Omega_{+-}\Omega_{-+}}{\Omega_{--}\Omega_{++}}\right)\right], & \\
+\frac{4}{\sqrt{\Omega_{++}\Omega_{--}}}\left[2\Pi\left(\frac{-\Omega_{-+}}{\Omega_{++}}, \frac{\Omega_{-+}\Omega_{+-}}{\Omega_{++}\Omega_{--}}\right) - K\left(\frac{\Omega_{-+}\Omega_{+-}}{\Omega_{++}\Omega_{--}}\right)\right], & 0 \le \omega < -\Delta + V, \\
\frac{2}{\sqrt{V/\Delta}}\left[\Omega_{--}\Pi\left(\frac{-\Omega_{-+}}{2V/\Delta}, \frac{-\Omega_{-+}\Omega_{+-}}{4V/\Delta}\right) + (1 - \Omega_{--})\cdot K\left(\frac{-\Omega_{-+}\Omega_{+-}}{4V/\Delta}\right)\right], & -\Delta + V < \omega < \Delta + V, \\
\frac{4}{\sqrt{-\Omega_{-+}\Omega_{+-}}}\left[K\left(\frac{4V/\Delta}{-\Omega_{-+}\Omega_{+-}}\right) - \Omega_{--}\Pi\left(\frac{2V/\Delta}{-\Omega_{-+}}, \frac{4V/\Delta}{-\Omega_{-+}\Omega_{+-}}\right)\right], & \Delta + V < \omega.
\end{cases}
\tag{A.7}
$$

In these equations $K(n)$ is the complete elliptic integral of the first kind and $\Pi(n, m)$ is the complete elliptic integral of the third kind. These results for DOS are presented in the Fig. 1. We note, that these expressions have logarithmic singularities at energies $\omega = \pm(\Delta + V)$ and discontinuous jumps at $\omega = \pm(\Delta - V)$, that replace the conventional square-root BCS divergence in the DOS at $\omega = \pm\Delta$.

## B Approximations in Green function

In matrix form, Green function from the Eq. (7) can be rewritten in the following way,

$$
G^{-1} = i\epsilon^{(1)}
$$
$$
-\begin{pmatrix}
\xi_{\mathbf{k}} & -i\lambda_{\text{so}}ke^{-i\phi_{\mathbf{k}}} + iV^{(1)} & -i\Delta^{(2)} & \Delta^{(1)} \\
i\lambda_{\text{so}}ke^{i\phi_{\mathbf{k}}} - iV^{(1)} & \xi_{\mathbf{k}} & -\Delta^{(1)} & -i\Delta^{(2)} \\
i\Delta^{(2)} & -\Delta^{(1)} & -\xi_{\mathbf{k}} & i\lambda_{\text{so}}ke^{i\phi_{\mathbf{k}}} + iV^{(1)} \\
\Delta^{(1)} & i\Delta^{(2)} & -i\lambda_{\text{so}}ke^{-i\phi_{\mathbf{k}}} - iV^{(1)} & -\xi_{\mathbf{k}}
\end{pmatrix}.
\tag{B.1}
$$

In order to separate interband and intraband terms, we make the rotation with the same matrix $U$ as defined in Eq. (A.1), $\tilde{G} = U^{\dagger}GU$,

$$
\tilde{G}^{-1} = i\epsilon^{(1)}
$$
$$
-\left(\begin{array}{cc|cc}
\xi_{\mathbf{k}} + \lambda_{so}k - V^{(1)}\cos\phi_{\mathbf{k}} & ie^{-i\phi_{\mathbf{k}}}\left(\Delta^{(1)} - \Delta^{(2)}\cos\phi_{\mathbf{k}}\right) & iV^{(1)}\sin\phi_{\mathbf{k}} & \Delta^{(2)}e^{-i\phi_{\mathbf{k}}}\sin\phi_{\mathbf{k}} \\
-ie^{i\phi_{\mathbf{k}}}\left(\Delta^{(1)} - \Delta^{(2)}\cos\phi_{\mathbf{k}}\right) & -\xi_{\mathbf{k}} - \lambda_{so}k - V^{(1)}\cos\phi_{\mathbf{k}} & \Delta^{(2)}e^{i\phi_{\mathbf{k}}}\sin\phi_{\mathbf{k}} & -iV^{(1)}\sin\phi_{\mathbf{k}} \\
\hline
-iV^{(1)}\sin\phi_{\mathbf{k}} & \Delta^{(2)}e^{-i\phi_{\mathbf{k}}}\sin\phi_{\mathbf{k}} & \xi_{\mathbf{k}} - \lambda_{so}k + V^{(1)}\cos\phi_{\mathbf{k}} & -ie^{-i\phi_{\mathbf{k}}}\left(\Delta^{(1)} + \Delta^{(2)}\cos\phi_{\mathbf{k}}\right) \\
\Delta^{(2)}e^{i\phi_{\mathbf{k}}}\sin\phi_{\mathbf{k}} & iV^{(1)}\sin\phi_{\mathbf{k}} & ie^{i\phi_{\mathbf{k}}}\left(\Delta^{(1)} + \Delta^{(2)}\cos\phi_{\mathbf{k}}\right) & -\xi + \lambda_{so}k + V^{(1)}\cos\phi_{\mathbf{k}}
\end{array}\right).
\tag{B.2}
$$

Now, interband terms are in off-diagonal blocks 2 by 2. Assuming the strong spin-orbit coupling: $\lambda_{\text{so}}k \gg V^{(1)}$ and $\lambda_{\text{so}}k \gg \Delta^{(2)}$, we can neglect interband terms which is equivalent to neglecting terms contributions of the order of $[V^{(1)}/(\lambda_{\text{so}}k_F)]^2$ and $[\Delta^{(2)}/(\lambda_{\text{so}}k_F)]^2$.

## C Derivation of a response kernel $Q_{\alpha\beta}$

In this appendix we present the derivation of the electromagnetic response kernel $Q_{\alpha\beta}$. To construct the kernel, we use the explicit expression for current operator that is given by the sum of three terms when written in the real space,

$$
\hat{\boldsymbol{j}} = \hat{\boldsymbol{j}}^{(\text{grad})} + \hat{\boldsymbol{j}}^{(\text{so})} + \hat{\boldsymbol{j}}^{(\text{dia})},
\tag{C.1}
$$

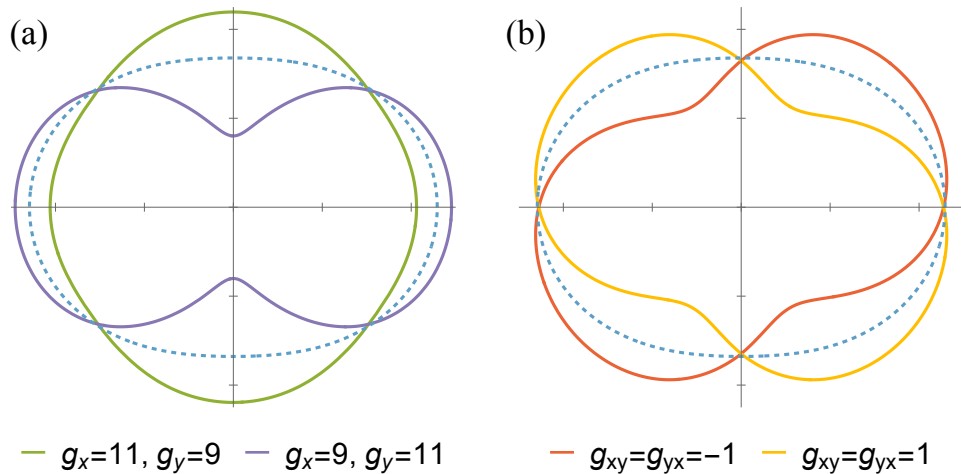

(a) (b)

$\quad g_x=11, g_y=9 \quad g_x=9, g_y=11 \qquad g_{xy}=g_{yx}=-1 \quad g_{xy}=g_{yx}=1$

Figure 8: Dependence of the normalized superfluid density $n_{s,xx}/n$ on the angle between the magnetic field and $x$-axis for different forms of $g$-tensor. Dashed blue line in both panels serves as a reference and corresponds to isotropic and diagonal $g$-tensor with $g_{xx} = g_{yy} = 10$ and $g_{xy} = g_{yx} = 0$. Panel (a) shows the effect of the anisotropy in the diagonal terms of $g$-tensor, when off-diagonal terms are vanishing. Panel (b) demonstrates that the off-diagonal terms in the $g$-tensor tilt principal symmetry axes. Radial divisions start at $n_{s,xx}/n = 0$ and are in increments of 0.2; $\mu_B H = 0.13\Delta_{00}$ and $\tau\Delta = 0.5$.

that represent the standard gradient contribution, spin-orbit contribution, and, finally the diamagnetic term,

$$\hat{\boldsymbol{j}}^{(\text{grad})} = \frac{ie}{2m}(\partial_{\boldsymbol{r}'} - \partial_{\boldsymbol{r}})|_{\boldsymbol{r}'\to\boldsymbol{r}}\psi^\dagger(\boldsymbol{r}',\tau)\psi(\boldsymbol{r},\tau), \tag{C.2}$$

$$\hat{\boldsymbol{j}}^{(\text{so})} = \lambda_{so}e\psi^\dagger(\boldsymbol{r},\tau)\begin{bmatrix} \sigma_y \\ -\sigma_x \end{bmatrix}\psi(\boldsymbol{r},\tau), \tag{C.3}$$

$$\hat{\boldsymbol{j}}^{(\text{dia})} = -\frac{e^2}{m}\psi^\dagger(\boldsymbol{r},\tau)\boldsymbol{A}(\boldsymbol{r})\psi(\boldsymbol{r},\tau). \tag{C.4}$$

Separating the diamagnetic contribution, that results in the term proportional to the total carrier density, $n$, the kernel $Q_{\alpha\beta}(\mathbf{q}, i\omega_n)$ can be expressed as follows

$$Q_{\alpha\beta}(\mathbf{q}, i\omega_n) = -\frac{ne^2}{m}\delta_{\alpha\beta} + \frac{1}{2}\int d\boldsymbol{r}\,e^{-i\boldsymbol{q}\boldsymbol{r}}\int_{-\beta}^{\beta} d\tau\,e^{i\omega_n\tau}$$
$$\times \left\langle (j_\alpha^{(\text{grad})}(\boldsymbol{r},\tau) + j_\alpha^{(\text{so})}(\boldsymbol{r},\tau))(j_\beta^{(\text{grad})}(0,0) + j_\beta^{(\text{so})}(0,0))\right\rangle, \tag{C.5}$$

where we indicated the coordinate and time dependence of operators. Substituting explicit expressions for electric current, rewriting the correlator using Green's function, and expressing the diamagnetic contribution as the product of Green's functions with $\Delta$ set to zero we obtain Eq. (13) in the main text. We note, that there is no vertex correction from disorder due to the following facts: 1) scattering potential is local; 2) the Green function $G(\boldsymbol{p})$ is an even function of $\boldsymbol{p}$. As a result, the vertex correction vanishes because it is proportional to the following integral: $\int_{\boldsymbol{p}} \boldsymbol{p}_\alpha G(\boldsymbol{p})G(\boldsymbol{p})$, which is equal to 0 since the integrand is an odd function of $\boldsymbol{p}$. More details on how vertex correction is calculated can be found in Refs. [44, 45], where disorder potential is not assumed to be local and as a result, the vertex correction is non-trivial.

# D  Anisotropic effects

## D.1  Superfluid density in the presence of magnetic anisotropy

In this Appendix we examine the different effects of magnetic anisotropy, encoded into $g$-tensor on the superfluid density, $n_{s,xx}$ (in the absence of order parameter suppression). In particular, we show the effect of the off-diagonal $g$-tensor and discuss potential for observing its effect experimentally.

We use Eq. (21) to plot the dependence of the superfluid density, $n_{s,xx}$, on the angle between a magnetic field and $x$-axis for different forms of $g$-tensor. First, we consider the case when off-diagonal elements of $g$-tensor are absent, $g_{xy} = g_{yx} = 0$ and diagonal terms are different, see Fig. 8(a). Note, that even in this case the superfluid density depends on the direction of the magnetic field – a property that stems from the spin-momentum locking. Upon including the anisotropy $g_{xx} \neq g_{yy}$, the dependence of $n_{s,xx}$ on the angle of the field becomes either more symmetric when $g_{xx} > g_{yy}$. In the opposite case, $g_{xx} < g_{yy}$, the dependence of $n_{s,xx}$ on the field angle develops more pronounced asymmetry and assumes an hourglass shape, with the smallest superfluid density corresponding to the case when magnetic field is parallel to the $y$-axis. Next, we explore the case when off-diagonal elements of $g$-tensor are present and diagonal are equal, see Fig.8(b). This type of anisotropy gives the dependence also an hourglass shape, and tilts its principal symmetry axes away from $x$ and $y$ directions.

The qualitative difference in the shape of $n_{s,xx}$ as a function of the angle may be used to detect and quantify the $g$-tensor anisotropy in the experiment. At the same time, addition of the off-diagonal elements in the $g$-tensor as fitting parameters may potentially result in the overfitting, especially if other six fitting parameters used in the main text are still present. Hence, in order to unambiguously determine the form of the $g$-tensor using this framework, it is desirable to fix at least some of the other material properties using different or additional experimental data, or directly measure off-diagonal components of the superfluid density as discussed in the main text.

## D.2  Generalization of spin-orbit coupling

In the main text we assumed that spin-orbit coupling is of a Rashba type. In this Appendix we show that our results are applicable not only to Rashba spin-orbit coupled system and can be generalized by a simple redefinition of the $g$-tensor. In particular, they can be also applied to the system that has only Dresselhaus spin-orbit coupling.

Rashba spin-orbit coupling has the following contribution to the Hamiltonian in momentum representation,

$$h_{\text{so}}^{(R)}(\boldsymbol{k}) = \lambda_{\text{so}} \begin{bmatrix} \sigma^x & \sigma^y \end{bmatrix} \begin{pmatrix} 0 & -1 \\ 1 & 0 \end{pmatrix} \begin{bmatrix} k_x \\ k_y \end{bmatrix}. \tag{D.1}$$

Instead of $2 \times 2$ antisymmetric tensor above, we consider more general case – an arbitrary orthogonal matrix $U$. Any orthogonal matrix $U$ can be expressed as:

$$U = U_{\text{rot}}^T \begin{pmatrix} 0 & -1 \\ 1 & 0 \end{pmatrix} \begin{pmatrix} d & 0 \\ 0 & 1 \end{pmatrix}, \tag{D.2}$$

where $U_{\text{rot}} = \begin{pmatrix} \cos\theta & -\sin\theta \\ \sin\theta & \cos\theta \end{pmatrix}$ is an arbitrary rotation matrix and $d = \pm 1$ corresponds to reflections. Then, we make the changes of variables,

$$\begin{bmatrix} \tilde{\sigma}_1 \\ \tilde{\sigma}_2 \end{bmatrix} = U_{\text{rot}} \begin{bmatrix} \sigma^x \\ \sigma^y \end{bmatrix}, \qquad \begin{bmatrix} \tilde{k}_x \\ \tilde{k}_y \end{bmatrix} = \begin{pmatrix} d & 0 \\ 0 & 1 \end{pmatrix} \begin{bmatrix} k_x \\ k_y \end{bmatrix}, \tag{D.3}$$

that corresponds to rotation in the spin space and potentially reflection in the momentum space. After this transformation, the Hamiltonian takes the same form as for the case of Rashba spin-orbit coupling in Eq. (D.1).

Let us now analyze how this change of variables affects other terms in the Hamiltonian. To begin with, we consider the Zeeman contribution,

$$H_Z = -\frac{1}{2}\mu_B \vec{\boldsymbol{\sigma}}^T \hat{g} \mathbf{H} \to -\frac{1}{2}\mu_B (\vec{\tilde{\sigma}})^T \hat{\tilde{g}} \mathbf{H}, \tag{D.4}$$

here, $\hat{\tilde{g}} = U_{\text{rot}}\hat{g}$. It is clear that the reflection in momentum space does not affect the kinetic term $H_k$ because it is quadratic in momentum, the same applies to the calculation of diagonal components of superfluid density tensor; off-diagonal components are multiplied by factor $d$ since they are proportional to $k_x k_y$. Thus, we have reduced the problem to the original one and all results for Rashba spin-orbit coupling are applicable to this generalization of spin-orbit coupling, the only change is the redefinition of $g$-tensor: $\hat{g} \to U_{\text{rot}}\hat{g}$. In particular, if we put $\theta = \pi/2$ in $U_{\text{rot}}$ and $d = -1$ in reflection matrix, we get the Dresselhaus spin-orbit coupling.

# E An effect of out-of-plane component of magnetic field on pair-breaking parameter

In Sec. 4.1 we studied pair-breaking in Al and found a linear contribution from the magnetic field to the pair-breaking parameter $\alpha$. Assuming that this linear contribution appears because of a small out-of-plane component of a magnetic field in the experimental measurements, let us estimate this component. The pair-breaking parameter for the case when a magnetic field is perpendicular to the layer is expressed as follows: $\alpha_\perp(H) = DeH$ [58, 59], while for the magnetic field parallel to the layer, it reads: $\alpha_\parallel(H) = D(eHd)^2/6$, where $D$ is the diffusion constant, $d$ is the thickness of the superconducting layer, $e$ is the charge of the electron, and $H$ is a magnetic field.

Assuming that pair-breaking is weak, $\alpha/\Delta_{00} \ll 1$, we can make the following approximation for resulting pair-braking parameter: $\alpha(H) \approx \alpha_\parallel(H_\parallel) + \alpha_\perp(H_\perp)$ where $H_\perp = H\sin\phi$ and $H_\parallel = H\cos\phi$ are out-of-plane and in-plane components of magnetic field and $\phi$ is an angle between a magnetic field and the plane. Then, taking into account Eq. (31) we obtain that

$$\frac{\Delta_{00}}{2}\zeta_1\left(\frac{H}{H_{\text{cr}}} + \frac{1-\zeta_1}{\zeta_1}\frac{H^2}{H_{\text{cr}}^2}\right) \approx DeH_\perp + \frac{D}{6}\left(eH_\parallel d\right)^2. \tag{E.1}$$

Matching linear and quadratic in $H$ terms with each other separately we get,

$$\frac{\Delta_{00}}{2}\zeta_1\frac{H}{H_{\text{cr}}} \approx DeH_\perp, \tag{E.2}$$

$$\frac{\Delta_{00}}{2}(1-\zeta_1)\frac{H^2}{H_{\text{cr}}^2} \approx \frac{D}{6}\left(eH_\parallel d\right)^2. \tag{E.3}$$

Taking the ratio of these equations we obtain

$$\frac{\zeta_1}{1-\zeta_1} \approx \frac{6}{\pi} \cdot \frac{\Phi_0}{\Phi} \cdot \frac{\sin\phi}{\cos^2\phi}, \tag{E.4}$$

where $\Phi_0$ is the magnetic flux quantum, $\Phi = H_{\text{cr}} \cdot d^2$ is the flux of the critical field through the area of $d^2$. Using the value of the fitting parameter $\alpha_1 = 0.207$ obtained in the main text and values $H_{\text{cr}} = 0.93$ T and $d = 15$ nm from Ref. [1] we estimate the misalignment angle $\phi$ as

$$\phi \approx \frac{H_\perp}{H_\parallel} \approx 0.01. \tag{E.5}$$



Figure 9: Results for renormalized parameters $\Delta^{(1)}$ (the first line), $\Delta^{(2)}$ (the second line) and $V^{(1)}$ (the third line) in the absence of order parameter supression: $\Delta = \Delta_{00}$. Plots (a-c) correspond to the real parts, plots (d-f) – to the imaginary part. The green color represents the magnetic field at which the system is in the regime with a gap, while the blue one represents the gapless regime.

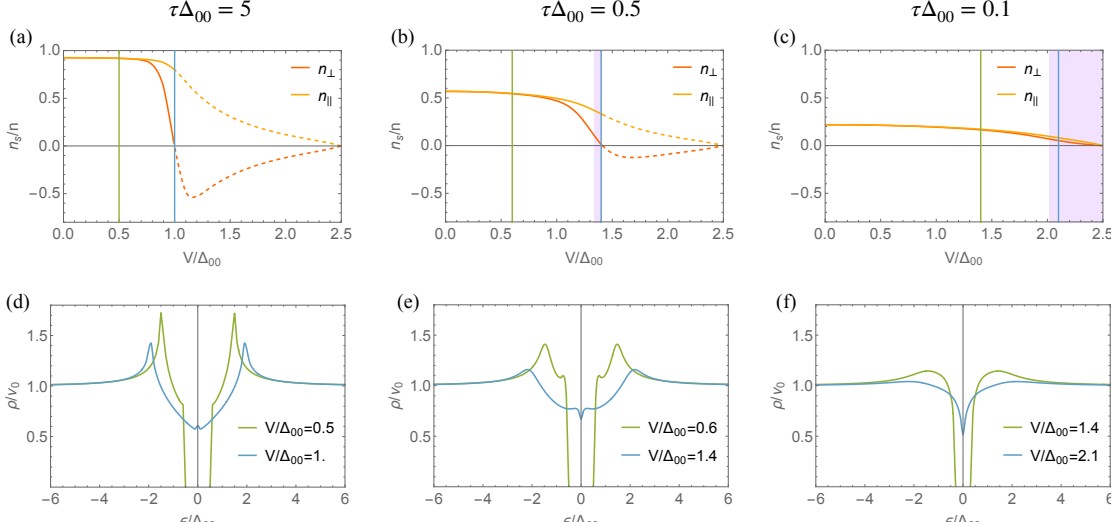

Figure 10: Results for superfluid density (top) and DOS (bottom) in the presence of order parameter suppression. For the calculation of superfluid density and order parameter suppression, the following temperature and critical magnetic field are used: $T/\Delta_{00} = 0.0564$, $V_{\mathrm{cr}} = 2.5\Delta_{00}$. Dashed lines correspond to the areas of magnetic field where current theory is not applicable (superfluid density becomes negative). Purple areas represent the regions of magnetic field where there is a gapless regime. Vertical lines in the plots from the upper panel show the magnetic field at which the DOS is plotted in the lower panel; the green line represents the magnetic field at which the system is in the regime with a gap, while the blue one represents the gapless regime.

We should notice that this estimate is applicable only when pair-breaking is weak which corresponds to the small magnetic fields: $H/H_{cr} \ll 1$. Experimentally Ref. [1] performed very careful field alignment with the out of plane component being constrained to less than 0.1% at considerable values of the magnetic field, that is an order of magnitude smaller compared to our estimate above. This discrepancy may be attributed to the fact that our estimate is applicable only at very small fields, also the systematic contributions that are beyond our theoretical model, such as field-dependence of inverse proximity effect, may play a role here.

## F Renormalization of the order parameter and magnetic field by disorder

In this Appendix, we present the dependencies of real and imaginary parts of parameters $\Delta^{(1)}$, $\Delta^{(2)}$, and $V^{(1)}$ on energy. The data is shown for three values of disorder strength: $\tau\Delta = 0.1$, $\tau\Delta = 0.5$, and $\tau\Delta = 5$, with each plot comparing the renormalized functions for two values of magnetic field that put the system into regime with the gap, and gapless regime.

Figure 9 demonstrates that for energies far away from zero, real part of gaps, $\Delta^{(1,2)}$ and renormalized magnetic field $V^{(1)}$ tend to their true values, and imaginary parts approach zero. At the same time, for energies of order $V$ and $\Delta$, renormalized functions strongly depend on the energy. In particular, all renormalized functions have features at the energies of the order $\pm|V \pm \Delta|$. For small disorder (large $\tau$) these features are sharp, and they smoothen out with increasing disorder strength.

Another notable feature is the symmetry properties of the real and imaginary part of the renormalized functions. First, for small values of $V$ when DOS has the gap, the $\operatorname{Im}\Delta^{(1)}(\epsilon)=0$ for $\epsilon$ that is sufficiently small. The same property also holds for imaginary parts of the $\operatorname{Im}\Delta^{(2)}(\epsilon)$ and $\operatorname{Im}V^{(1)}(\epsilon)$. Second, the renormalized s-wave gap component, $\Delta^{(1)}(\epsilon)$ has even in energy real and odd in energy imaginary parts.

In contrast, the induced triplet component has odd in energy real and even in energy imaginary parts, highlighting that the presence of disorder induces *odd frequency triplet component* of the order parameter [69]. The magnitude of the triplet component is increasing with disorder strength, and for large disorder it becomes in some energy intervals even larger that the order parameter in the superconductor. At the same time, in this regime both real and imaginary contributions to $\Delta^{(2)}(\epsilon)$ are of comparable magnitude, intuitively suggesting that disorder-induced triplet pairs are short lived.

# G   Superfluid density and DOS in the presence of order parameter suppression

In this Appendix, we present the plots for superfluid density and DOS in the presence of order parameter suppression. Taking values of the temperature $T = 0.0564\Delta_{00}$ and critical field $V_c = 2.5\Delta_{00}$ we show resulting superfluid density and DOS in Fig. 10. Data in this figure is qualitatively similar to the case where the order parameter suppression is not taken into account, see Fig. 2 in the main text. The only differences are that the superfluid density vanishes at the critical field of superconductor, as expected and the boundaries of the gapless regimes move to smaller values of magnetic field, which can also be observed in the plots in Fig. 3.

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
