# Peer review of "Proximity-induced gapless superconductivity in two-dimensional Rashba semiconductor in magnetic field"

_SciPost Physics, doi:SciPost Phys. 16, 115 (2024)_

## Round 1 · Referee Report · Anonymous (Referee 1) · 2024-2-6

Report

In this manuscript the authors perform a detailed study of the emergence of a Bogoliubov Fermi surface in a proximitized 2D electron gas with Rashba spin-orbit coupling when it is placed in an external in-plane magnetic field. Most importantly, they include the effect of non-magnetic disorder solving self-consistently for the self-energy and show that the magnetic field magnitude range for which a gapless superconducting phase exists increases with larger disorder. Based on self-consistent solutions, they provide predictions for both the density of states as well as the superfluid density for different disorder strengths and magnetic field magnitudes. They also provide a detailed prescription for fitting the experimental data from recent work, enabling extraction of such parameters as anisotropic g-factors.

This work touches upon a very relevant and timely topic of Bogoliubov Fermi surfaces resulting from external magnetic fields or the supercurrent flow. The authors extended the current literature studies by self-consistent treatment of disorder and explained their calculation procedure in detail, which gives this manuscript a great pedagogical value. They also carefully explain the experimental fitting procedure, which I greatly appreciate. Therefore, the study certainly has a great value, and I could potentially see it published in SciPost Physics. One doubt I have is that according to the acceptance criteria, the work accepted in this journal is supposed to be a groundbreaking discovery or should open a new research direction, which this paper is not necessarily satisfying. Nevertheless, before making the final decision, I would like the authors to discuss the following issues:

1. The authors did not consider the parent superconductor that is the source of the proximity effect in their self-consistent calculation. While I agree that the inverse proximity effect on the parent superconductor may be of lesser importance, I am wondering if presence of such a superconductor would increase the robustness of the gapless phase beyond the very narrow range of magnetic fields in the clean systems. Would there be some sort of “pinning” effect that would stabilize the proximity-induced order parameter, especially if the magnetic field is smaller than what would close the gap in the parent superconductor? Doing the full calculation is probably too much work, but I would be grateful if the authors could provide some analysis of this issue.

2. Could the authors make some comparisons to the situation in which the topological Dirac surface state is proximitized, as was the case in the experiment in Science 374, 1381-1385 (2021)? This could be especially relevant since that reference claims to have observed Bogoliubov Fermi surface due to in-plane magnetic field. Do the authors expect that the impact of disorder is stronger in Rashba 2DEG as compared to the topological surface state?

3. Did the authors consider the possibility of the appearance of the in-plane vortices? In some superconductors like NbSe2 such vortices can appear at magnetic fields as low as 100 mT, smaller than the field that would lead to closing of the superconducting gap. How would the presence of such vortices impact the conclusions of the current analysis?

Minor points:
- Ref. 60 and 61 seem to refer to the same publication. Please remove the duplicate references.
- There are some issues with capitalization in reference titles, especially when considering chemical compounds, like in Ref. 62

---

## Round 1 · Referee Report · Anonymous (Referee 2) · 2024-2-7

Strengths

The main strength is the clear and thorough investigation of a theoretically important and experimentally relevant problem.
The paper is very well written and presents a consistent study of the gapless state tied to the Bogoliubov Fermi surfaces as well as the superfluid density in the spin-orbit coupled proximitized two-dimensional system subject to a parallel magnetic field.

Report

Dear editor,
Please find the report below.

Authors investigate the density of states and the superfluid density of the proximitized two-dimensional electron gas (2DEG) with Rashba spin-orbit coupling in the presence of the in-plane magnetic field. The boundary between the systems is assumed to be fully transparent. The authors carefully investigated the dependence of the evolution of the gapless superconductivity with the disorder in the 2DEG and the strength of the magnetic field.
The order parameter is fixed by the superconducting substrate assumed to be in a dirty limit.
The gap in the substrate is determined by numerically solving the Eilenberger equations in conjunction with the self-consistency equation. The density of states is obtained by solving the Gorkov equations using the iteration technique.
The disorder in the 2DEG is shown to stabilize the gapless state which is shown to emerge from the Bogliubov Fermi surfaces appearing in the clean system as the Zeeman splitting reaches the superconducting gap. At elevated magnetic fields the negative superfluid density is interpreted as indicative of an instability. Authors list few options as of how such instability could have been resolved.
Finally the authors perform the detailed six-parameter fit of their theoretical results to the existing experimental data. The fit is shown to be very good.

I would like to recommend this high quality work for publication.
Still there are few technical issues I would encourage the authors to address in order to improve and clarify the presentation.
These are given below in the section "Requested changes"

In summary, once the changes requested below are addressed the paper should be published.

The referee

Requested changes

1) Author stress the anisotropy of the g-tensor. I wonder of why this anisotropy is so significant, and what could be the origin of it.
2) i.e. attempt perhaps should read e.g. attempts
3) Is the full transparency of the interface a sufficient condition to set the two order parameters being equal in the superconductor and 2DEG? Could the author clarify when such assumption is a fair description of the proximity effect in the current setting?
4) On a technical side. Appendix B is meant to clarify how the condition of strong spin orbit compared to the gap and Zeeman splitting is used to invert the matrix needed to get the Green function. I am puzzled on which terms exactly are neglected in the Eq. (B2). The elements that are at first row and third column (1-3 element) connects the states differing in energy by 2 \lambda k and indeed can be neglected. Similarly 2-4 element can be ignored. Is that what is done or some other elements have been dropped as well? The element (1-2) connects states that might get degenerate for some momenta, so it seems one cannot neglect them. Am I missing something?
5) I am puzzled by the regime $g_{yx} \gg g_{xy}$. I thought that this tensor is symmetric. One can, for instance relate it to the second derivative of a suitable free energy with respect to the components of the field. Please clarify.
6) The departing model for the superconductor with quadratic field dependence of the Abrikosov-Gorkov parameter assumes the superconductor is thinner than the coherence length and not just than the penetration depth, as otherwise the order parameter acquires a noticeable spatial dependence. Is that the case for s system studied?
7) In the calculation of the DOS authors used the cutoff and the grid of 0.1\Delta in real frequency. Do one need a cutoff for DOS: the calculation is done per frequency. And is the grid fine enough to resolve the subtly features such as van Hove singularities?
8) Author show the reduction of the superfluid density at zero field by the disorder. Could the author compare it against a well-known results (the spin-orbit should not change it, I believe) ?
9) In the detailed fit to the data in Eq. (30) apart from the quadratic dependence of the depairing on the field the linear piece is needed as well. My question is could one interpret is as a signature of the Maki result for the bulk Hc2 in dirty limit?

---

## Round 2 · Author Response

Dear Editor,

Thank you for arranging reviews of our paper.

We are grateful to both referees for providing constructive comments that helped us to improve the publication. While the second referee report recommends the publication of our work, the first referee report casts doubts on the impact of our work, and also asks important questions about relation to the previous work and other systems. In the resubmitted version of the paper we address comments from both reports, which helped us to improve the presentation in our paper.

Regarding the impact of our work, we would like to point out that the hunt for “a groundbreaking discovery” lead to reproducibility crisis in the field of Majorana physics. We do not claim that our work has such a discovery, yet we do believe that our paper opens “a new research direction”, by establishing an novel methodology of quantitative characterization of hybrid superconductor-semiconductor structure. Our work builds the theory, that, supplemented by the sensitive experimental measurements of the cavity resonance shift, allows to quantitatively extract the in-situ parameters of material and leads to verifiable experimental predictions. We strongly believe that our approach can be applied to a broad family of proximitized heterostructures and enables inference of the Hamiltonian parameters purely from experimental data.

We hope that the resubmitted manuscript addresses referee comments in the satisfactory way and is suitable for publication at the SciPost Physics.

Serafim
On behalf of the Authors

---

## Round 2 · List of Changes

Reply to Anonymous Report 2

We are grateful to the anonymous Referee for their positive characterization of our work and for constructive comments. Below we explain our answers to the requested changes:

1) Author stress the anisotropy of the g-tensor. I wonder of why this anisotropy is so significant, and what could be the origin of it.

The motivation to consider anisotropic g-tensor is dictated by the fact that large spin-orbit materials often have strongly anisotropic g-factors. The intuition is that spin-orbit coupling mixes the spin and orbital motion of electron and thus it is also capable of yielding direction-dependent response to the applied magnetic field. A well-studied case, which is also of particular relevance to this work, is that of zincblende quantum wells. In this case, off-diagonal components of the g-tensor arise in proportion both to the strength of the Dresselhaus spin-orbit coupling and to the strength of asymmetry in the quantum well. In experimentally relevant cases, this can easily give rise to large relative changes in the g-factor of 50% or more.

In order to provide a better motivation for the anisotropic g-tensor, we expanded the first paragraph in the section II E with the motivation behind considering anisotropic g-tensor.

Until now we considered the magnetic field pointing along y-direction and did not take into account possible g-tensor anisotropy. However, $g$-tensor anisotropy is generically expected in the presence of spin-orbit coupling. Indeed, in the relevant case of spin-orbit coupled asymmetric quantum wells, in-plane $g$-factor anisotropy is known to be a large effect [49-51].

We also expanded the references to include the relevant experimental literature.

2) i.e. attempt perhaps should read e.g. attempts

We fixed the typo.

3) Is the full transparency of the interface a sufficient condition to set the two order parameters being equal in the superconductor and 2DEG? Could the author clarify when such assumption is a fair description of the proximity effect in the current setting?

Although we did not perform the microscopic modeling of the semiconductor-superconductor interface, we believe that transparency of the interface leads to the induced order parameter being close in magnitude. In our work we assume that order parameters are directly equal to each other, in order to avoid introducing additional unknown or fitting parameters into the model. In practice, it is possible to measure the induced gap in the 2DEG via the tunneling barrier, thereby directly inferring the magnitude of the order parameter. The difference between induced order parameter and order parameter in the parent superconductor can be easily added to our theoretical model as an additional parameter, but it does not change the qualitative physics.

In order to clarify this point in the paper, we modified the last paragraph of the Section II A, so it reads:
 The absence of inverse proximity effect along with the assumption of transparent interface between semiconductor and superconductor (we note that our model can be easily extended to include suppression of the induced order parameter due to imperfect interface transparency, in practice it is also possible to characterize induced order parameter in the 2DEG via tunneling measurements) allows us to treat the 2DEG with the induced order parameter that is entirely determined by depairing intrinsic to superconductor.

We also expanded the last section of the paper:

 Bringing additional ingredients into our theoretical model, that already relied on a six-parameter fitting for describing experimental data, would most likely require additional experimental probes. In particular, it is easy to relax the assumption that the induced order parameter in the 2DEG is equal to its value in the superconductor, and relying on the tunneling measurements use more accurate relation.

4) On a technical side. Appendix B is meant to clarify how the condition of strong spin orbit compared to the gap and Zeeman splitting is used to invert the matrix needed to get the Green function. I am puzzled on which terms exactly are neglected in the Eq. (B2). The elements that are at first row and third column (1-3 element) connects the states differing in energy by 2 \lambda k and indeed can be neglected. Similarly 2-4 element can be ignored. Is that what is done or some other elements have been dropped as well? The element (1-2) connects states that might get degenerate for some momenta, so it seems one cannot neglect them. Am I missing something?

Spin-orbit coupling results in the splitting of energy bands. In our treatment, we neglect interband terms and retain only intraband terms. In the matrix form in Eq. (B2), the upper left 2 by 2 block corresponds to one Rashba-split band, and the lower right 2 by 2 block corresponds to the other band Rashba band with the reversed direction of spin-momentum locking. Therefore, the elements with indices i=1, j=3 and i=2, j=4 are interband terms and are neglected; meanwhile, the element i=1 and j=2 is intraband and is fully taken into account in our treatment.

In order to clarify this point we added vertical bars into the equation B2.

5) I am puzzled by the regime g_yx>>g_xy. I thought that this tensor is symmetric. One can, for instance relate it to the second derivative of a suitable free energy with respect to the components of the field. Please clarify.

Since the g-tensor couples spin to magnetic field, it is not constrained to be symmetric in the general case (unlike the superfluid density tensor, which is symmetric and can be derived as the second derivative of free energy with respect to the components of the vector potential). In fact, there are several known examples of the systems with asymmetric g-tensor, such as low symmetry quantum wells. This is not the case for InAs, nevertheless, we generalize our theory so it can be potentially applied for other 2DEG where asymmetry actually takes place. We added the following passage in the manuscript in the last paragraph of the Section II E with the corresponding citation [52]: 
 In spin-3/2 hole systems within low-symmetry quantum wells (e.g., GaAs), the $g$-tensor can exhibit asymmetry due to the interaction between the p-like character of hole wave functions and asymmetric band-edge profiles, alongside specific crystallographic orientations [52].

6) The departing model for the superconductor with quadratic field dependence of the Abrikosov-Gorkov parameter assumes the superconductor is thinner than the coherence length and not just than the penetration depth, as otherwise the order parameter acquires a noticeable spatial dependence. Is that the case for system studied?

In experimentally relevant systems, the thickness of the superconductor is much smaller than the superconducting coherence length. Taking Ref. [1] as an exemplary case, the superconducting film is at most 15 nm thick, and the dirty-limit coherence length, accounting both for the measured critical temperature and the Al mean free path, is 70 nm.

7) In the calculation of the DOS authors used the cutoff and the grid of 0.1\Delta in real frequency. Do one need a cutoff for DOS: the calculation is done per frequency. And is the grid fine enough to resolve the subtly features such as van Hove singularities?

Indeed we agree with the referee that word “cutoff” is misleading. By "cutoff" for the DOS, we refer to the energy value up to which the DOS has been numerically calculated. Beyond this values, no distinctive features of the DOS are observed; it remains nearly constant. Concerning the van Hove singularities, they exhibit a very weak, logarithmic nature even in the absence of disorder. We believe that the introduction of disorder effectively suppresses these singularities, leading to the emergence of local maxima instead. The numerical data clearly show that all dependencies are smooth, with no evidence of singularities. This effect is particularly pronounced in cases of strong disorder, where even the maxima are weak.

In order to avoid potential confusion we modified the Section IIIA so the sentences

As a cutoff for energy, we use the following value: $\epsilon_\text{max}=6 \Delta$, so, $\epsilon \in [-\epsilon_\text{max},\epsilon_\text{max}]$

The cutoff for the energy is $\epsilon_\text{max}=40\cdot 2\pi T$.

now read:

We note that iterative procedure can be performed independently at each energy separately. As a maximum value of the considered energy we use: $\epsilon_\text{max}=6 \Delta$, so, $\epsilon \in [-\epsilon_\text{max},\epsilon_\text{max}]$, since no additional feature were observed beyond this range.

The maximal value for the considered energy is $\epsilon_\text{max}=40\cdot 2\pi T$

8) Author show the reduction of the superfluid density at zero field by the disorder. Could the author compare it against a well-known results (the spin-orbit should not change it, I believe)?

We compared our results with well-known analytical formula from [47]:
\frac{n_s}{n}=\pi T \sum_{\omega_n} \frac{\Delta^2}{\left(\omega_n^2+\Delta^2\right)^{3 / 2}\left(1+\frac{1}{2 \tau \sqrt{\omega_n^2+\Delta^2}}\right)} for three values of disorder for which superfluid density is plotted in Fig. 2: \tau =0.1: \frac{n_s}{n}=0.22, \tau =0.5: \frac{n_s}{n}=0.57, \tau =5: \frac{n_s}{n}=0.93. It is clear that they coincide with ones on the plots. The corresponding comment with citation were added to the manuscript: “(zero-field values of superfluid density are in agreement with well-known analytical results [47])” to the first paragraph of the Section III B.

9) In the detailed fit to the data in Eq. (30) apart from the quadratic dependence of the depairing on the field the linear piece is needed as well. My question is could one interpret is as a signature of the Maki result for the bulk Hc2 in dirty limit?

We are grateful to the referee for the useful suggestion. While we believe that our film is far away from the bulk limit, indeed, according to the theory of Maki there is an expected crossover between two different behaviors: linear in field pair breaking (for a bulk sample), and quadratic (for a thin film). We believe that in our case the film is thin enough so the dominant contribution from in-plane magnetic field is quadratic. However, if the out-of-plane component is present, the thickness of film is not relevant for it, and the contribution from out-of-plane component can be treated as for a bulk sample, which gives linear contribution to the pair-breaking parameter. Hence, in response to the referee comment we expanded the discussion with the mention of Maki’s result (Ref [59] in Appendix E) and modified the third paragraph in section II F: to read “in thin film (the thickness of film is much smaller than coherence length)”.

Reply to Anonymous Report 1 1. The authors did not consider the parent superconductor that is the source of the proximity effect in their self-consistent calculation. While I agree that the inverse proximity effect on the parent superconductor may be of lesser importance, I am wondering if presence of such a superconductor would increase the robustness of the gapless phase beyond the very narrow range of magnetic fields in the clean systems. Would there be some sort of “pinning” effect that would stabilize the proximity-induced order parameter, especially if the magnetic field is smaller than what would close the gap in the parent superconductor? Doing the full calculation is probably too much work, but I would be grateful if the authors could provide some analysis of this issue.

We agree with the intuition provided by the referee. On the one hand, the negative superfluid density suggests instability of the state with gapless Bogoliubov Fermi surfaces. On the other hand, the 2DEG cannot be considered alone and it is coupled to the big “parent superconductor” that provides the pairing mechanism. Due to the latter fact, we do not expect that this instability would completely destroy the pairing in the whole system. As we speculate in Section IIIC the instability may induce the weak modulation of the order parameter that may suffice to remove the large density of states at the Fermi surface, without strongly renormalizing the band structure and properties away from it. Such state may behave still very similar to the state with Bogoliubov Fermi surfaces with respect to many experimental probes, and in this aspect the big superconductor “saves”/“stabilizes” the system from major reconstruction of the band structure.

In order to highlight this intuition we modified the discussion in Section V that now reads:

Another set of questions that remain open is the eventual fate of the gapless proximity-induced superconducting state in 2DEG at sufficiently weak disorder and in the clean limit [1]. While the presence of superconducting film suggests that this instability will not destroy pairing in the system, it may cause reconstruction of the low energy band structure leading to reduced density of states.

  1. Could the authors make some comparisons to the situation in which the topological Dirac surface state is proximitized, as was the case in the experiment in Science 374, 1381-1385 (2021)? This could be especially relevant since that reference claims to have observed Bogoliubov Fermi surface due to in-plane magnetic field. Do the authors expect that the impact of disorder is stronger in Rashba 2DEG as compared to the topological surface state?

We are grateful to the referee for bringing this paper to our attention. This paper is certainly relevant to our work, although there are important distinctions. Specifically, in our theoretical model we assume that Zeeman terms give dominant energy contribution from the in-plane magnetic field. In contrast, the Science paper estimates the orbital effect (i.e. contribution from the phase shift induced by in plane field) to be two orders of magnitude stronger compared to Zeeman. While proper treatment of orbital contribution in presence of disorder is beyond our work, we believe that qualitatively interplay of disorder with Zeeman and orbital contributions from magnetic field are similar. Thus, our expectations are that disorder is similar to the case of Zeeman. Namely, we expect that disorder may help stability of the Bogoliubov’s Fermi surfaces also in the case of surface states of topological insulator, and it will broaden the features in the density of states, while keeping the qualitative form of all features intact. We also not note that some presence of disorder is necessary for the quasiparticle interference experiments as reported in the Science paper, and our mechanism of stabilization does not require large disorder (since we operate in regime when , that implies ), and thus such disorder may well be present on the surface of the topological insulator.

In response to referee comments we now cite the paper in the introduction and in Section V when discussing the orbital effects.

The introduction was expanded as follows:

In addition to 2DEG proximitized by conventional s-wave superconductors, the physics of Bogoliubov Fermi surfaces and interplay between disorder, spin-orbit coupling and superconductivity is actively studied in other material systems. In particular, Bogoliubov Fermi surfaces in presence of in-plane magnetic field were probed by the scanning tunneling microscopy experiments performed on the proximitized surface states of topological insulator [25].

More broadly, our theoretical model with some modifications will be applicable to a much broader family of materials, such as hybrid semiconductor [36] and topological insulator [37,38] nanowires or proximitized surface states [25], Germanium based 2DEGs [39], ferromagnetic hybrids [11], and two-dimensional materials with strong spin orbit coupling such as transition metal dichalcogenides.

Section V was expanded as follows:

Although our model was able to describe the experimental data and generate predictions, a number of questions remains open. First, while our theoretical model incorporates disorder compared to earlier theoretical studies, it still relies on a number of approximations. In particular, it may be desirable to relax the assumption of the perfectly transparent interface between 2DEG and superconductor, incorporate orbital effect of the magnetic field and inverse proximity effect -- phenomena related to more realistic description of the motion of electrons between 2DEG and superconductor [65]. Incorporation of orbital effects may be particularly important for Germanium that has small value of $g$-factor and also for surface states of topological insulator proximitized by the bulk superconductor [25] where orbital contribution dominates the physics.

  1. Did the authors consider the possibility of the appearance of the in-plane vortices? In some superconductors like NbSe2 such vortices can appear at magnetic fields as low as 100 mT, smaller than the field that would lead to closing of the superconducting gap. How would the presence of such vortices impact the conclusions of the current analysis?

We are grateful to the referee for bringing up a potentially important issue. Our work is predominantly theoretical, we build the model for fitting the experimental data, and it is crucial that theory uses correct assumptions.

Vortex occurrence is common for superconductors of type II subject to magnetic field. In our theoretical model we assumed the absence of vortices and homogeneous penetration of magnetic filed into superconductor because the thickness of superconductor is much smaller than the penetration depth. Experimentally, the absence of vortices was checked via the control experiment that had only Al-film and no 2DEG. In this case the superfluid density was in perfect agreement with Al depairing theory, that did not take into account any vortices, thus providing experimental confirmation of their absence.

In order to highlight the assumption about the absence of vortices, we modified Section V with a footnote [65] that explicitly mentions this assumption and the fact that vortices may lead to a very different physics:

[65] Note, that we assume the homogeneous penetration of magnetic field into superconducting film, implying the absence of vortices. This is crucial ingredient for applicability of our model, and it may be verified in the control experiment where the response of superconducting film without 2DEG is compared to standard depairing theory.

---

## Editorial Decision

published